# Exploring the potential of artificial intelligence in individualized cognitive training: A systematic review

**Maxime Adolphe**[1,2,3]*, **Marion Pech**[1☯], **Masataka Sawayama**[4☯], **Denis Maurel**[2‡], **Alexandra Delmas**[2‡], **Pierre-Yves Oudeyer**[1‡], **Hélène Sauzeon**[1,3‡]

**1** Flowers, Inria, Bordeaux-Sud-Ouest, France, **2** R&D, onepoint, Bordeaux, France, **3** Bordeaux Population Health, INSERM, Bordeaux, France, **4** Graduate School of Information Science and Technology, University of Tokyo, Tokyo, Japan

☯ These authors contributed equally to this work.
‡ DM, AD, PYO, and HS also contributed equally to this work.

* maxime.adolphe@inria.fr

**Data availability statement:** All the data and information necessary to reproduce the findings of this study are fully presented in the

## Abstract

To tackle the challenge of responders heterogeneity, Cognitive Training (CT) research currently leverages AI Techniques for providing individualized curriculum rather than one-size-fits-all designs of curriculum. Our systematic review explored these new generations of adaptive methods in computerized CT and analyzed their outcomes in terms of learning mechanics (intra-training performance) and effectiveness (near, far and everyday life transfer effects of CT). A search up to June 2023 with multiple databases selected 19 computerized CT studies using AI techniques for individualized training. After outlining the AI-based individualization approach, this work analyzed CT setting (content, dose, etc.), targeted population, intra-training performance tracking, and pre-post-CT effects. Half of selected studies employed a macro-adaptive approach mostly for multiple-cognitive domain training while the other half used a micro-adaptive approach with various techniques, especially for single-cognitive domain training. Two studies emphasized the favorable influence on CT effectiveness, while five underscored its capacity to enhance the training experience by boosting motivation, engagement, and offering diverse learning pathways. Methodological differences across studies and weaknesses in their design (no control group, small sample, etc.) were observed. Despite promising results in this new research avenue, more research is needed to fully understand and empirically support individualized techniques in cognitive training.

## 1 Introduction

The repetitive and prolonged practice of specific cognitive activities, more often called "Cognitive Training" (CT), is an umbrella concept with multiple dimensions and multiple issues. First of all, in the field of aging or neurocognitive rehabilitation, the hope of finding non-drug and non-invasive interventions is a path to be favored in first-line clinical care. Indeed, the presence of neurocognitive disorders or declines has a major impact on the comfort of life

manuscript. No additional data are available for sharing.

**Funding:** SH: Contract ESR2020-23 - Systèmes numériques personnalisés pour l'entraînement de l'attention.

of the persons, and can lead to a decrease in autonomy, or even a slide towards a pathological condition [1]. Thus, many researchers have mobilized their workforce in the design of training or cognitive rehabilitation programs for older adults, for Mild Cognitive Impairment (MCI) patients [2–5], Alzheimer's patients [6–8], Parkinson's patients [9,10], or any patient with Acquired Brain Injury (ABI) [11,12]. Second, outside of these health issues, research on CT is growing to meet the needs of performance enhancement in certain activities: sports performance [13], academic performance [14] or even professional performance [15,16]. Lastly, alongside the difficulties related to the restoration and enhancement of performance, CT constitutes a fundamental realm of exploration encompassing the study of learning mechanisms, their evolution, and their neural associations [17]. Given the expansive nature of CT, which cover a diverse range of cognitive skills, interventions, as well as social and commercial implications, an open letter written by 70 researchers in 2014 brought attention to the challenge of inadequate compelling evidence in this complex and multifaceted field. In 2016, a response from 111 researchers acknowledged areas needing improvement while emphasizing the continued promise of various research directions. Subsequently, despite ongoing debate fueled by studies both supporting [18,19] and challenging [20–22] CT, research in this domain has witnessed heightened activity. Notably, the volume of publications on PubMed in 2016 surpassed the cumulative studies conducted in preceding years [23], indicating a surge in scholarly interest and engagement. In the perspective of contributing to the improvement of this field, our Systematic Review (SR) explores adaptive methods of customizing the training program to each individual.

Specifically, our work addresses the central challenge of managing variability in response to CT, which includes both inter-individual and intra-individual differences. To address this variability, we notably focus on strategies that leverage artificial intelligence (AI), a rapidly advancing field increasingly applied to CT. AI methods, offer the potential to revolutionize CT by enabling a range of adaptations. These include personalizing the training path by tailoring difficulty and content to participants' specific needs [24], predicting potential dropouts [25], or providing real-time, tailored feedback through tools like conversational agents [26]. As such, our review explores these AI-driven approaches, focusing on how they can effectively mitigate the challenges posed by variability in CT outcomes and improve overall participant engagement and success.

As described in [27–29], prior cognitive performance, age, and education is a non exhaustive set of factors that influence the magnitude of the impact of the interventions. The compensation effect (greater CT-related improvement of participants with lower prior performance [30,31]) and the magnification effect (greater CT-related improvement of participants with higher prior performance) are observed in many studies [27,32,33]. Thus, in order to maximize the likelihood of program response, many interventions proposed adapting the difficulty and content to participants. This adaptation can be implemented manually, before or during the program, by the designer or the health professional [34]. Utilizing prior knowledge of the participant's progression and performance during training, these methods can also be implemented automatically (e.g [35]. Classically, automatic approaches are based on a staircase procedure where the difficulty increases if the participant successfully completes several activities in a row and decreases otherwise [36,37]. Originating in the field of psychophysics [38–40], the use of staircase strategies for training has the advantage of bringing the participant to his maximum capacity and pushing him to exceed it. However, even if these so-called "adaptive" procedures are easy to deploy in computerized CT systems, they lack flexibility and responsiveness in their ability to individualize the procedure. First, they do not take account the whole learning trajectory followed by the participant (only some of the previous activities are considered for the calculation of the future activity) [24,25]. This

suggests that a participant who has temporarily dropped to a lower level of difficulty due to factors such as fatigue or inattention, will be presented with the same task as another participant who has reached their true limit of learning, and will have to invest an equivalent amount of time to regain their previous maximum level. Secondly, this strategy poses challenges in managing a substantial number of parameters concurrently as it becomes complex to infer the progression of difficulty when multiple parameters are altered simultaneously [41]. Thirdly, staircase strategies result in a limit around which participants oscillate until improvement is observed. As a consequence, since participants consistently encounter similar stimuli near the threshold, this pattern can generate a perception of repetition that may be demotivating, discouraging, and not conducive to effective training. While certain programs [42] have suggested incorporating adaptive steps to update task difficulty, they still exhibit limited parameter involvement in controlling the difficulty. Consequently, the training activities' space remains underutilized for the learner, restricting the range of learning opportunities for progress. Finally, the inflexible structure inherent in the unique trajectory design dictated by the staircase strategy obstructs the integration of the abundant knowledge and theories available from diverse fields like education sciences and psychology. For instance, a notable drawback is its inability to accommodate various signals from learners, such as physiological measurements (EMG, EEG), posture, or interaction data (like clicks), which can be valuable for tailoring the choice of educational activities and gaining insights into how learners react to the curriculum they receive. Considering the limitations outlined above, this systematic review aims to emphasize novel approaches for tailoring interventions to individual participants' needs. Thus, the interventions incorporated into this review will be labeled as "Individualized Computerized CT," contrasting them with the majority of self-proclaimed "Adaptive Computerized CT" to support the aspiration of providing genuine personalization to each participant.

Beneath the inquiry into the variability of CT responses lies the fundamental question of how to assess the effectiveness of these interventions. Traditionally, CT effectiveness is evaluated in terms of the extent of impact with a short-term spectrum corresponding to local effectiveness (improvement in performance on tasks similar to those trained, i.e., near effect) and a broad spectrum corresponding to global effectiveness (improvement in performance on tasks not similar to those trained but involving common cognitive mechanisms and functions, i.e., far effect). This range of impact is expressed in terms of Near and Far transfer (NFT) [43]. The NFT effects are generally assessed using cognitive batteries [44–48] and allow the evolution of the participant's performance after training to be quantified. Research in this domain frequently concludes after establishing efficacy, without delving into the ecological transfer of training i.e the practical influence of training on real-life tasks [20]. This gray area can be attributed by the fact that the ecological validity of CT is difficult to objectify, except with the use of assessments with a more ecological content or questionnaires in which participants are asked to self-report the improvements perceived in real life. These tasks or questionnaires often have methodological limitations (ecological content validity, and subjective bias [49]). In addition to these considerations of effectiveness measures, many SRs or reviews raise weaknesses in the level of evidence provided by the studies (e.g., [43]. These weaknesses are related to the study design (i.e presence of a control group [50], randomization of group assignment, blindness of researchers and participants, sample size, etc.) [43] and the design of the interventions (nature and type of training task, dosage, etc.) [19]. Echoing the reproducibility crisis of science, it is observed that some studies showed significant effects of CT, while others are unable to reproduce these results. Among the salient factors identified, the lack of standardization of the content used is highlighted by recent SRs [51]. Consequently, this review

will give particular attention to the methodological decisions and the resulting conclusions, striving to provide a thorough depiction of the field's status.

To the best of our knowledge, no SR has been proposed to identify the new generation of individualized CT and to analyze their impact in terms of near or far effectiveness. In compliance with the PRISMA standards, as illustrated in the flow diagram (Fig 1) and the checklist provided in the supporting information, this study aims to concentrate on interventions that offer more adaptable strategies, facilitating enhanced individualization of content. We are particularly interested in CT proposing either automatic individualization of multimedia content or of the difficulty of the task. Inclusion criteria for this review necessitate that strategies facilitate the tailoring of interventions to individuals or representative groups. Such strategies should enable the generation of individualized and optimized learning trajectories for each learner. Hence, this criterion for inclusion implies the utilization of automation strategies spanning different levels of intelligence, notably those grounded in artificial intelligence.

## 1.1 Research question

The current systematic review of the literature first investigates the individualization strategies employed in computerized CT tools (Sects descriptive results, Q1–Q2). Secondly, it questioned the motivations of researchers to produce this type of strategy, i.e., specific individualization goals targeted by the strategy (Sect Q1–Q2). Finally, it examined the effectiveness of the included studies in light of the quality of the evidence provided, i.e., study design and statistical power (Sects Q3, Q4, Q5). The ultimate aim was therefore to establish an inventory of existing flexible adaptive strategies and their level of maturity to serve the field of CT.

## 1.2 Background

The development of adaptive methods in CT is mainly fed by two main research fields, i.e., the field of computerized CT and the field of intelligent tutoring systems (ITS) even if the contribution of the latter one is larger to those of the former (see for reviews, [52,53].

**1.2.1 Insights from adaptive computerized CT research.** This line of research has mainly contributed to exploring staircase methods for CT. Often, these methods consisted of the execution of graded exercises, whose difficulty increases gradually according to a set of predefined rules, considering the results the trainees achieve. Frequently, predefined rules are derived from expert knowledge. For instance, the exercises are typically structured hierarchically according to difficulty levels, and the progression between levels is primarily determined by predefined thresholds, often set at 70% of correct answers for each level of exercises. Hence, the staircase methods consist of a unique trajectory design of CT program, involving that all trainees follow a single path although at different speeds or with a different number of attempts. Several computerized CT systems for various CT purposes are based on this design of program personalization [54], such as Brainer [55], Neurotracker® [56], Reha-Com®. [57], CogniPlus®. [58], HappyNeuron Pro®. [59], Erica [60], the Padua Rehabilitation Tool (PRT) [61], MS Rehab [62], Cogni-Track [63] and CogniFit Personal Coach® [64]. In the majority of investigations that have contrasted adaptive strategies of this design with conventional approaches, a consistent finding has been the enhanced CT outcomes associated with adaptive strategies (as evidenced by studies like [35,36,65]). Nevertheless, contrasting results have emerged in certain studies, exemplified by [37], which did not detect any advantages under adaptive conditions. Notably, this particular study implemented adaptive adjustments between sessions rather than within the same session, which may account for the disparity in outcomes.

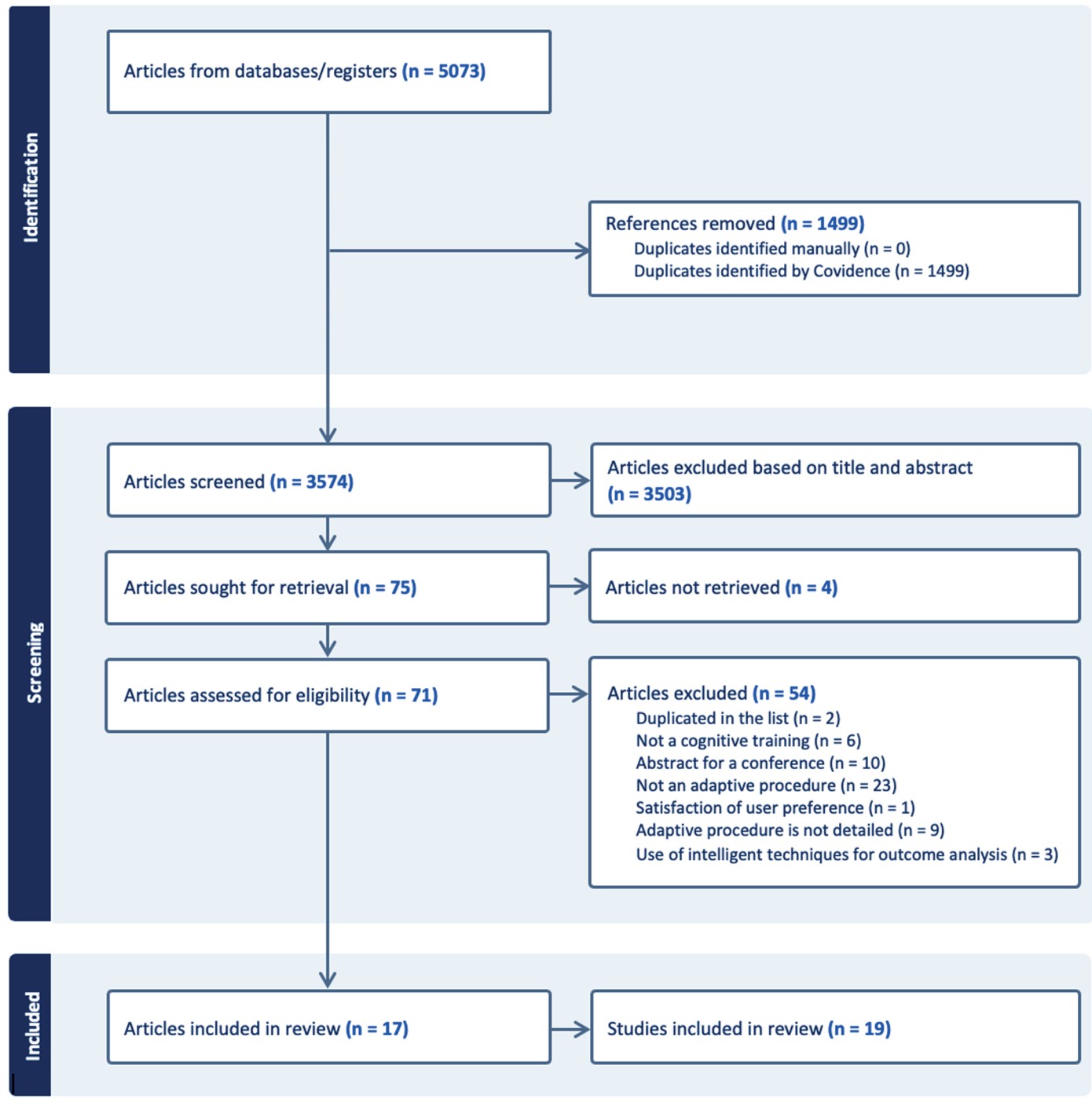

**Fig 1. PRISMA Flow chart.**

https://doi.org/10.1371/journal.pcbi.0316860.g001

From a more distant standpoint, recent SR highlighted the promising findings from CT studies comparing gamified contents to non-gamified ones as a result of the critical role of trainee 's motivation and engagement in the individualization of CT [66,67]. Taken together,

these overall results supported the added value of personalization of CT for fostering their outcomes. However, as mentioned above, the staircase methods have some limitations and are not really adaptive due to its single trajectory design, i.e, the system adapts the CT in the same manner for all trainees instead to specifically adapt the program to the trainee performance by creating a specific path into the program. Due to this strong limitation, most of existing computerized CT systems (e.g., HappyNeuron Pro®, Erica, MS-Rehab®) include a manual calibration for defining the initial level of exercise at the beginning of the CT and the successive tweaks of difficulty level across the CT (often done by the clinicians). However, as the trained tasks involve a significant number of parameters to determine the level of difficulty, manual calibrations become increasingly complex and numerous.

**1.2.2 Insights from ITS research.** Educational sciences have widely contributed to demonstrate that factors intrinsic to the learner (such as prior knowledge, emotional load, mental load or motivation) and extrinsic factors (such as all the variables related to the instructional design), are mediators of the efficiency of the learning functions. By nature, the effectiveness of CT is no exception to this observation and responds to similar factors. Hence, it seems natural to ask how effective instructional methods from the educational sciences can be transferred to the field of CT, and more particularly those providing an individualization of learning. Learning theories indicate that learning requires an appropriately sized "mismatch" – a gap between the cognitive capacity and the requirements of the external task that the cognitive system must adapt to in order to improve performance [68]. As a result, the evidence-based assets of individualized learning over one-size-fits-all educational approaches are today well documented [69,70]. ITS offers a framework for the automated creation of curricula tailored to individual students. While there are multiple methods available to enrich and personalize educational content with ITS for each learner, the majority of systems are organized around three primary components [52]. Firstly, there is the aspect of adapting to the instructional source, which refers to what the system will tailor, including aspects like the learner's learning style [71,72], existing knowledge [73], or preferences [74]. Secondly, there is the target of adaptive instruction, specifying what aspects will undergo adaptation. This could involve the content of the instruction [71] or the manner of presentation [75]. Thirdly, the adaptive component functions as the intermediary, creating a pathway between the first two components. It dictates how to adapt a target to a source, which can be achieved through diverse methods. This last component, also called the tutoring module, is the engine generating a curriculum of training activities for learners in ITS. Adaptive feedback, hint, and recommendation-generating, navigation of the learning path, and presenting adaptive educational content constitute the core of this component [76]. The content's adapting to the learner's needs is the most relevant tutoring dimension of ITS for the individualization purpose into a CT program. In order to tailor content to individual learners, numerous ITS draw from the concepts of the zone of proximal development (ZPD) [77] and the state of Flow [78]. These concepts are closely tied to the Goldilocks effect [79,80], wherein learning is optimal in tasks that strike a balance between simplicity and excessive challenge. Following them, many ITS aim to offer the learner pedagogical activities that are neither too difficult nor too easy with regard to their abilities, so that they can be engaged and progress in their acquisitions without being anxious or bored during the process. ITS can also suggest activities that may be challenging for the learner to solve independently, but become manageable with the assistance of hints or guidance from the teacher. According to this ZPD principle, the tutoring component classically integrates a performance threshold principle for exercise difficulty shift (often chosen around 70%) to maintain an average optimal learning trajectory [79]. Several signals or performance dimensions can be used to guide the generation of a curriculum:

some ITS are interested in using an optimal emotional level [81] or learning progress [82,83] or both [84].

### 1.2.3 Sorting Keys of AI techniques for content adapting to learner's capabilities.

The first criterion for categorizing strategies in ITS is based on the level of flexibility they provide. One category, often referred to as 'linear designs' [85], predicts an optimal path through the activity space, guiding all learners along the same trajectory. Although learners may progress at different paces or require varying numbers of attempts, they all follow this predetermined route. In contrast, the second category, known as "branched-path designs," allows each learner to follow a personalized path tailored to their individual needs. In this review, we refer to this as the "Individualized" approach. As a result, learners experience diverse trajectories, which may be linear, non-linear (with jumps and backtracking), or hybrid in nature, making these designs truly adaptive to each learner. In summary, this first distinction divides the strategies into two groups based on the degree of adaptability they offer.

The second key criterion for categorizing strategies relates to the scale at which adaptability occurs, distinguishing between macro-adaptive and micro-adaptive procedures. As illustrated in Fig 2, macro-adaptive procedures operate at a broader level, adjusting the sequence and selection of tasks. For instance, in a multi-domain CT, the system might prioritize certain types of tasks or cognitive domains for each learner depending on their performance across different areas, ensuring that the task sequence is tailored to their evolving strengths and weaknesses. In contrast, micro-adaptive procedures make adjustments at a finer, more granular level. These systems adapt the difficulty of tasks in real time, modifying specific task parameters after each learner interaction. For example, if a learner responds correctly to a trial, the system may increase the task difficulty slightly in the next step, whereas an incorrect response might prompt the system to decrease the complexity of the upcoming task. This ensures that the learner is consistently challenged at an appropriate level, maintaining engagement and optimizing the learning experience.

The third and final criterion pertains to the specific techniques used for adaptation. A variety of methods have been employed to tailor content to each learner's needs, and these can be broadly grouped into several categories and sub-categories, often applied individually or in combination [53,86]. These categories include rule-based systems, probabilistic approaches, and learning-based algorithms. Each technique presents unique strengths regarding flexibility, scalability, and responsiveness, enabling systems to dynamically adjust to learners' progress and performance in various ways:

- Condition-action rules-based reasoning traditionally refers to rule-based decisions (if X, then Y) that determined the outcome of adaptive instruction. Rules are set by the instructor prior to the learning process (e.g., rule-based expert system or semantic rule-based reasoning). In the context of CT, this category would encompass the staircase procedure as introduced previously.
- Machine learning techniques involve the use of algorithms and statistical models to enable computer systems to learn from data and improve their performance on a task without being explicitly programmed (see [87] for an introduction). In the context of ITS, machine learning techniques can be used to individualize the learning experience for each student by leveraging data collected during interactions with the system. The strategies can operate in two different modes: one is an incremental approach, where the model evolves during direct interactions with users, and the other is an offline method, which includes data collection, model development without real-time interaction, and its application to learners

thereafter. While the strategies are presented as distinct techniques, it is important to note that many of these concepts are interconnected and not mutually exclusive:

- Data mining refer to a set of techniques used to extract insights and knowledge from large datasets such as student interactions with the system or demographic data. These techniques involve analyzing the data to identify patterns and relationships that can be used to personalize the learning experience for each student. The extracted features can then be combined with decision-making modules to adapt the learning path and provide targeted support and guidance to the student. One example of a widely used data mining technique in ITS is clustering [12]. This method enables the identification of different groups of students based on their learning profile, needs, and preferences. By clustering students, ITS can create tailored learning paths that address the specific needs of each group, leading to more effective and efficient learning outcomes.

- Probabilistic modeling and Bayesian networks refer to a set of techniques that rely on graphical model to encode probabilistic relationships between variables of interest. A key advantage of using them is that their structure is ideal for combining prior knowledge, which is often in causal form, with observed data. Into an ITS, prior knowledge consists of a stereotyped model based on the learner's goals, tasks, and interests, while observed data is extracted from the interaction between the learner and the environment. Bayesian techniques can also be used when data is missing, a common problem in the learning sciences.

- Artificial neural networks and deep learning (DL) techniques are a set of techniques inspired by the structure and function of the human brain and are designed to learn from large datasets of student interactions with the system. In ITS, they can be used to model student behavior and performance, predict future outcomes, and adapt the learning experience to the individual needs of each student.

- Reinforcement learning (RL) is a type of machine learning in which an agent learns to make decisions in an environment by receiving feedback in the form of rewards or punishments. In the context of ITS, the RL agent can serve as an instructor and receive a reward based on the effectiveness of its pedagogical approach towards the student (see [88] for a review). Numerous algorithms have been developed to tackle this challenge. One common strategy involves maintaining a tabular record of how effective a specific pedagogical activity is, quantified by the cumulative rewards it garners, when employed with a student possessing a particular skill level. Through an iterative process of proposing various activities, the agent seeks to determine the optimal actions that maximize its overall reward (see Q-learning algorithm in [24] for an example). Another approach to address this challenge draws an analogy to a casino scenario featuring multiple slot machines. Within this metaphor, critical questions center on the selection of the most effective 'slot machines,' their optimal utilization frequency, and the establishment of a suitable sequence. In the educational context, these metaphorical 'slot machines' represent different pedagogical activities, and their success is gauged by the extent of knowledge acquisition by the student. To tackle the 'exploration-exploitation dilemma' inherent in this context, various techniques are employed such as multi-armed bandit algorithms (see [82] for an example).

- Natural language processing focuses on the interaction between computers and humans through natural language, including tasks such as text classification, sentiment analysis, and machine translation. ITS can use techniques such as text classification and sentiment analysis to understand students' written or spoken responses, enabling individualized feedback (see [89] for a review).

- Evolutionary algorithms are a family of optimization algorithms that are inspired by the process of natural selection and evolution to solve complex problems. By treating the potential solutions as a population of individuals possessing diverse traits, these algorithms employ a fitness function in conjunction with an evolutionary process to deduce the optimal solution (see [90] for a review). In the ITS literature, these techniques have been employed in various ways such as learner performance prediction or design of learning environments.

**1.2.4 Evaluation of AI techniques.** The evaluation methods of individualized techniques into ITS are of two kinds, either formal or empirical (for review; see [91]). Formal validations consist essentially in testing the system with simulations using learners' models for assessing the ITS behaviors in order to compare two or several AI techniques. Empirical validations are multiple-ways in terms of expected outcomes or study designs. The judgment criteria can be qualitative (i.e., experts or learners' feedback, learner experience questionnaires, etc.) or quantitative (i.e., learning performance, level of activities performed, etc.), or both. They can be based only on training phase (interaction data) or include pre- and post-training measurement, or both. Ensuring the validity of scientific research, whether validated formally or empirically, hinges upon the accessibility of both the dataset and the employed model. It

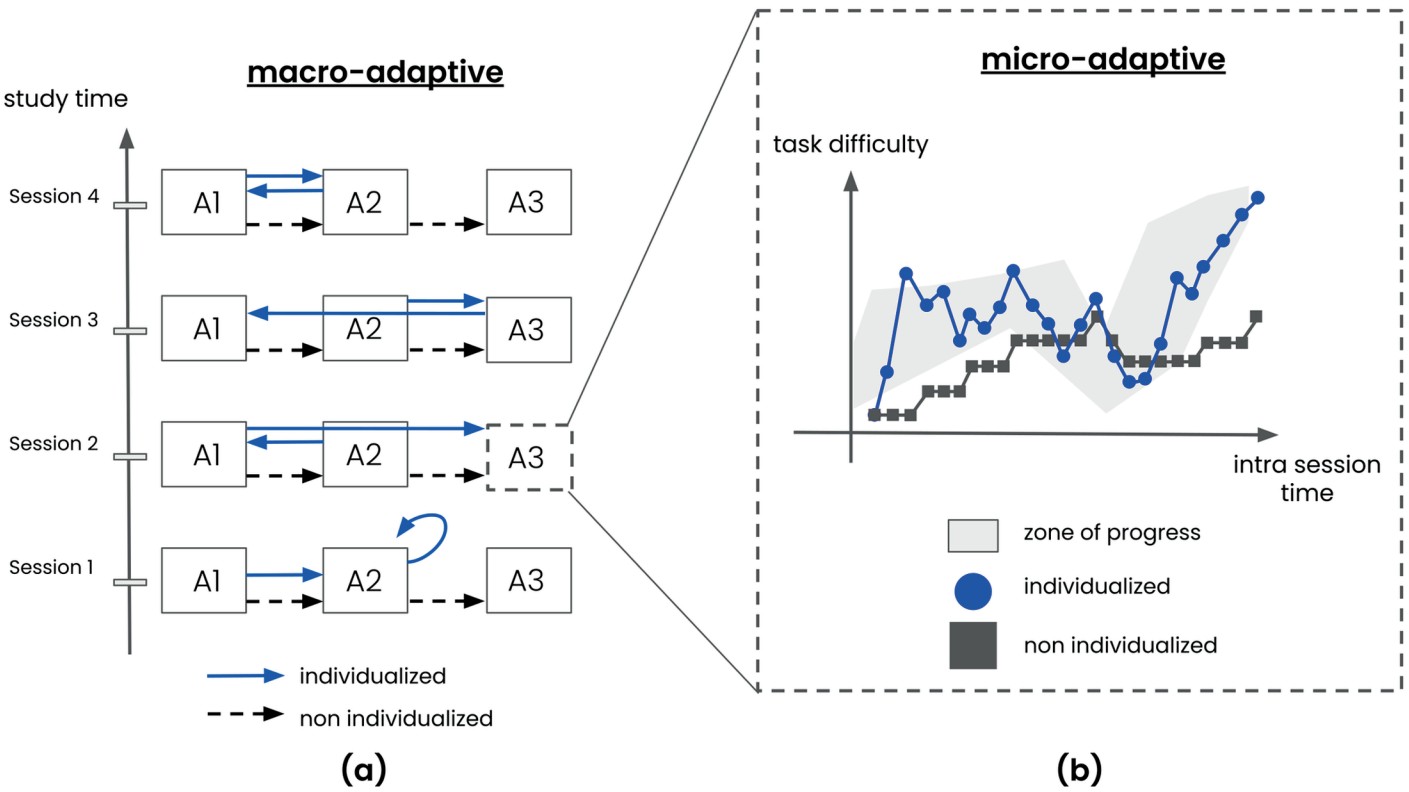

**Fig 2. Illustration of macro and micro-adaptive strategies.**
(a): Macro-adaptive strategy exemplified by two trajectories within a CT program (unique trajectory or individualized design) spanning sessions 1 to 4, each session offering three potential cognitive tasks (A1 to A3). Arrows depict task order for each session. Non individualized trajectory always propose same order A1, A2 and A3 while individualized path adapts the trajectory according to training objectives. (b): Micro-adaptive strategy demonstrated by two trajectories of task difficulty adjustments specifically for task A3 during session 2. The non-individualized trajectory relies on a staircase approach that falls short of identifying the optimal zone of progress when contrasted with the individualized procedure, which proves to be a more suitable fit.

https://doi.org/10.1371/journal.pcbi.0316860.g002

is worth noting that a notable factor contributing to the reproducibility crisis is the limited access to comprehensive research materials. The study design varies from feasibility or pilot study (e.g. prototype testing with few users) to Randomized Controlled Trial (large sample of individuals, control group, pre-and post-training measurement). The gold standard for evidence-based adaptive ITS is to compare it with a control condition often consisting of unique trajectory design (e.g staircase procedure) on qualitative and quantitative measurements taking place before, during and after the training and providing insights on NFT effects of the training.

**1.2.5 Operationalized research questions.** Pertaining to the central inquiry of this work - "Do the emerging generations of individualized strategies hold promise for computer-based cognitive training?" - five specific operational research questions were formulated as follows:

1. What AI Techniques have been employed in computerized CT, and what are the underlying research motivations driving their utilization?
2. What are the domains of CT for which adaptive techniques have been designed?
3. What populations are targeted and what are the characteristics of the CT settings?
4. How effective are individualized techniques in empirical CT studies? What effects are reported (NFT learning effects)?Are the effects dependent on characteristics of CT settings?
5. What Type of validation have been conducted for these new generations of computerized CT?

## 2 Material and method

A pre-established protocol was formulated and officially registered with PROSPERO (registration number: 2021 CRD42021241515). The checklist of the Preferred Reporting Items for Systematic Reviews and Meta-Analyses (PRISMA) was applied to guide the systematic review process (PRISMA). The COVIDENCE tool was also used to manage and organize the work.

### 2.1 Databases and search query

The initial database searches were conducted between February and April 2023 using the following electronic databases to conduct the study: PsycInfo, Medline, ETHOS, EMBASE, IBSS, PubMed, IEEE Xplore, ACM Digital Li-brary, Springer, Taylor Francis, Scopus, Education Resources Information Center (ERIC), ScienceDirect/Elsevier and EBSCO. In addition to the database searches, a hand search of relevant journals and gray literature were also conducted to ensure all relevant works were included in the review. According to the research question, we used the following query: TITLE-ABS-KEY ("Cognitive training") AND ALL ("Machine Learning" OR "reinforcement learning" OR "personalized*" OR "individualized" OR "intelligent tutoring system") AND NOT TITLE-ABS-KEY("Transcranial direct current stimulation").

### 2.2 Eligibility criteria

In this SR, we included all computerized CT studying individualized training that allows for differentiated learning paths in terms of content (type of exercises). No restrictions were set regarding the publication date, but the included studies had to be in English. Furthermore, no age or population criterion was used. Table 1 presents details of inclusion and exclusion criteria.

**Table 1. Inclusion and exclusion criteria.**

| Inclusion criteria | Exclusion criteria |
|---|---|
| The adaptive procedure relies on individual training performance and/or training experience. | The personalization only consists in satisfying user preference (e.g visual features, content type, gaming component) without adapting the learning path (e.g [92]) |
| The adaptive procedure is being tested on a dataset of CT, and the strategy is thoroughly described. | The adaptive procedure adheres to a "unique trajectory design" (e.g similar to staircase algorithms [93]) |
| Machine learning techniques are used to predict participants' behavior (adherence, success, emotional state, skill level) or to directly compute the optimal next activity | The adaptive procedure is not detailed (e.g [94]) |
| Machine learning techniques are used on training data from CT results, ECG, EEG, MRI, fMRI, wearable sensor data, and longitudinal training experience measures. | The intervention involves a form of neuromodulation (e.g., tDCS) |
| Neurofeedback and machine learning techniques are used for individualized programs | Intelligent techniques are used as tools for a better outcome analysis (e.g., effect size analysis) (e.g [95]) |
| Individualized techniques encompass both online strategies (where the participant engages while the model is developed) and offline strategies (involving data collection to build the model with no access to data during interactions) | Not a CT (e.g [96]) |
| English writing | Non-peer-reviewed papers, opinion pieces, or abstract conference papers |

## 2.3 Screening and selection method

The screening phase was conducted on articles until February 2023. In total, 5073 papers were found, as presented in Fig 1. All duplicates were removed, which reduced the results to 3574. Papers were selected through an iterative process of filtering. According to our search strategy (inclusion/exclusion criteria), studies were first filtered on titles and abstracts resulting in 71 articles to go through to the next stage of full-article review. All the screening process was carefully evaluated by two authors. When there was uncertainty or disagreement among the reviewers, consensus was reached through discussion. If no consensus was found, a third review was designed for the final decision. The full-text review of the remaining papers results in 17 papers with 19 studies included for the systematic review. The main reasons for the exclusions are reported in the PRISMA flowchart (Fig 1).

## 2.4 Data extraction

To answer our five research questions, four coding sheets were developed for extracting the searched information.

To address Q1 and Q2, information regarding the AI approach (macro-, micro-adaptive or both) and the AI techniques used, as well as the targeted cognitive domains of CT were collected in (Table 2). For the AI techniques, 8 families were distinguished : condition-action rules-based reasoning, probabilistic models or bayesian networks (e.g Kalman Filters (KF), Hidden Markov Models (HMM)), data mining (e.g Regression, Clustering), neural networks or deep learning (e.g multi-layered perceptron (MLP), convolutional neural networks (CNN), Long Short Term Memory (LSTM)), reinforcement learning (e.g Q-learning, Actor-critic), natural language processing, evolutionary algorithms and recommendation systems. Regarding cognitive domains, global CTs (multiple cognitive domains) were distinguished from

**Table 2. Overall descriptive results, AI techniques and Cognitive domains of CT for the selected studies.**

| Study | Date | Title | AI approach and AI Techniques | Cognitive domain of CT | Study design | Population Type and Age | Sample Size |
|---|---|---|---|---|---|---|---|
| García-Rudolph and Gibert [12] | 2014 | A data mining approach to identify cognitive NeuroRehabilitation Range in Traumatic Brain Injury patients | Macro-adaptive learning Decision tree objectives: Prediction of the optimal neurorehabilitation range | Multiple domains Attention, memory, language, executive functions | Non-randomised controlled trial | Clinical sample Age = [18–68] | n = 327 |
| Fermé et al. [97] | 2020 | AI-Rehab: A Framework for AI Driven Neurorehabilitation Training - The Profiling Challenge | Micro and macro adaptive Data mining and belief revision engines objectives: Participant profiling | Multiple domains Attention, Memory (semantic, episodic), language (understanding), reasoning (categorization) and problem-solving (maze, navigation task). | Feasibility study (study protocol) | NA | NA |
| Xu et al. [98] | 2018 | Personalized Serious Games for Cognitive Intervention with Lifelog Visual Analytics | Macro adaptive Deep learning and clustering techniques Personalization of game content with lifelog visual content | Multiple domains Attention, memory, visuo-spatial and executive functions | Individual randomized trial (crossover study) | Non clinical sample Age = 63.7 ± 7 | n = 26 |
| Reidy et al. [99] | 2020 | Facial Electromyography-based Adaptive Virtual Reality Gaming for Cognitive Training | Micro adaptive Data mining and machine learning EMG data preprocessing and affect classification | Multiple domains Memory (episodic), executive and problem-solving functions | Non-randomised controlled trial (crossover study) | Non clinical sample Age = [60–100] | n = 6 |
| Kitakoshi et al. (a) [100] | 2015 | Cognitive Training System for Dementia Prevention Using Memory Game Based on the Concept of Human-Agent Interaction | Micro adaptive Reinforcement learning (bucket brigade algorithm) Difficulty level adjustment and break offering system | Specific domain Memory (episodic) | Non-randomised controlled trial (crossover study) | Non clinical sample Age = [70–90] | n = 6 |
| Kitakoshi et al. (b) [101] | 2020 | A Study on Coordination of Exercise Difficulty in Cognitive Training System for Older Adults, study-1 | Micro adaptive Reinforcement learning (bucket brigade algorithm) Difficulty level adjustment | Specific domain Memory (episodic) | Non-randomised controlled trial (crossover study) | Non clinical sample Age = [70–90] | n = 5 |
| Kitakoshi et al. (b) [101] | 2020 | A Study on Coordination of Exercise Difficulty in Cognitive Training System for Older Adults - study-2 | Micro adaptive Reinforcement learning (bucket brigade algorithm) Difficulty level adjustment | Specific domain Memory (episodic) | Non-randomised controlled trial (crossover study) | Non clinical sample Age = 79.2 ± ND | n = 5 |

ND = Not Documented; NA = Not Applicable

**Table 2.** (Continued)

| Study | Date | Title | AI approach and AI Techniques | Cognitive domain of CT | Study design | Population Type and Age | Sample Size |
|---|---|---|---|---|---|---|---|
| Rathmayaka et al. [102] | 2021 | Cognitive Rehabilitation based Personalized Solution for Dementia PAtients using Reinforcement Learning | Micro adaptive Reinforcement learning (Q-learning) Difficulty level adjustment | Multiple domains Attention, memory, language, executive functions | Non-randomised controlled trial | Clinical sample Age = 76.4 ± ND | n = 56 |
| Shen and Xu [103] | 2020 | Research on children's cognitive development for learning disabilities using recommendation method | Macro adaptive Recommendation system (collaborative filtering) Proposition of a curriculum based on the similarity between children performances and preferences | Multiple domains Attention, memory, language, executive function (flexibility), reasoning | Individual randomized controlled trial | Non clinical sample Age = [10–11] | n = 30 |
| Sandeep et al. [104] | 2020 | Application of Machine Learning Models for Tracking Participant Skills in Cognitive Training - study-1 | Micro adaptive Machine learning and deep learning (Hidden Markov Model, Kalman filters, LSTM) Prediction of performance evolution through training | Specific domain (Working) Memory | Feasibility study (framework description - secondary data analysis) | Non clinical sample Age = 19.87 ± 2.32 | n = 262 (Dataset) |
| Sandeep et al. [104] | 2020 | Application of Machine Learning Models for Tracking Participant Skills in Cognitive Training - study-2 | Micro adaptive Machine learning and deep learning (Hidden Markov Model, Kalman filters, LSTM) Prediction of performance evolution through training | (Working) Memory | Feasibility study (framework description + secondary data analysis) | Non clinical sample Age = 19.79 ± 1.87 | n = 177 (Dataset) |
| Wilms [105] | 2011 | Using artificial intelligence to control and adapt level of difficulty in computed-based cognitive therapy | Micro adaptive Reinforcement learning (Actor-critic method) Difficulty level adjustment | Specific domain (Visual) Attention | Non-comparative Study (case study) | Clinical sample Age = 49 | n = 1 |
| Solana et al. [106] | 2014 | Intelligent Therapy Assistant (ITA) for cognitive rehabilitation in patients with acquired brain injury | Macro adaptive Clustering Definition of a cognitive impairment profile | Multiple domains Attention, memory, executive functions | Non-randomized controlled trial | Clinical sample Age = [16–55] | n = 582 |
| Zini et al. [24] | 2022 | Adaptive cognitive training with reinforcement learning | Micro adaptive Reinforcement learning (Q-learning) Difficulty level adjustment | Specific domain (Working) memory | Individual randomized controlled trial | Non clinical sample Age = 23.93 ± 2.29 | n = 20 |

ND = Not Documented; NA = Not Applicable

*(Continued)*

**Table 2.** (Continued)

| Study | Date | Title | AI approach and AI Techniques | Cognitive domain of CT | Study design | Population Type and Age | Sample Size |
|---|---|---|---|---|---|---|---|
| Zedda et al. [107] | 2022 | Towards Adaptation of Humanoid Robot Behaviour in Serious Game Scenarios using Reinforcement Learning | Micro adaptive Reinforcement learning (Q-learning) Robot's behavior personalization | Specific domain Attention (visual attention and working memory) | Non-randomised controlled trial (crossover study) | Non clinical sample Age = [28–45] | n = 3 |
| Eun et al. [108] | 2022 | Development and Evaluation of an Artificial Intelligence–Based Cognitive Exercise Game: A Pilot Study | Micro adaptive Deep learning (LSTM) Difficulty level adjustment | Multi domain : Physical training and cognitive training (attention, logic, response time, memory) | Non-comparative Study | Non clinical sample Age = [60–90] | n = 37 |
| Tsiakas et al. [109] | 2018 | Task Engagement as Personalization Feedback for Socially-Assistive Robots and Cognitive Training | Micro adaptive Reinforcement learning (Q-learning) Difficulty level adjustment | Specific domain Working memory and sequencing | Feasibility study (framework description - secondary data analysis) | Non clinical sample Age = NA | n = 69 (Dataset) |
| Book et al. [110] | 2022 | Individualised computerised cognitive training for community-dwelling people with mild cognitive impairment: study protocol of a completely virtual, randomised, controlled trial | Micro adaptive Machine learning (logistic regression) Prediction of performance evolution through training | Multiple domains Information processing speed Speed memory span Short term memory Logical reasoning | Feasibility study (study protocol) | Clinical sample Age = NA | n = 100 (Objective) |
| Singh et al. [25] | 2022 | Deep learning-based predictions of older adults' adherence to cognitive training to support training efficacy | Micro adaptive Deep learning (CNN, LSTM) Adherence prediction | Multiple domains Memory Attention Spatial processing, Task-switching, Reasoning, Problem-solving | Feasibility study (framework description, secondary data analysis) | Non clinical sample Age = 72.16 ± 5.5 | n = 18 (Dataset) |

ND = Not Documented; NA = Not Applicable

specific CTs addressing a single cognitive domain. We used the categorization of cognitive functions traditionally used in psychology, as follows: perception (visual, auditory, spatial, etc.), attention (selective, sustained, divided components), learning and memory (working, semantic, episodic, procedural), language (production and understanding), executive functions (inhibition, updating, and cognitive flexibility) and reasoning and problem solving (categorization, generalization, deductive and inductive inference, predictive and evaluative judgment).

For response to Q3, Table 3 aimed to collect descriptive data for each selected study in terms of population included, sample size, characteristics of CT design (content, dose, location). In addition, this sheet was also dedicated to Q4 as it relates to the effectiveness of AI based individualized computerized CTs according to several judgment criteria (intra-training performance, pre/post training effect, near/far effect, etc) (Table 3). In order to address Q4, we also developed a meticulously crafted scale to assess the presence of significant features that contribute to substantiating the effectiveness of the intervention. Indeed, as elucidated in [111], CT interventions must incorporate significant supplementary elements to demonstrate their effectiveness. Therefore, the proposed scale assigns a rating ranging from 0 to 3 for various dimensions, including information related to dosage and location, intra-training performance measures, subjective evaluation, pre-post comparisons, quality of the cognitive evaluation employed, and follow-up assessment. By summing the scores for all items, each study was assigned a grade ranging from 0 to 11.

Finally, Tables 4 and 5 collected information for a SIGN analysis [112] to assess the quality of study design in the field of individualized computerized CT. The SIGN ratings estimates the strength of available evidence provided by a study, based on the methodological design and the evaluation of possible biases. Regarding study designs, we considered various options as outlined in the SIGN guidelines. We included experimental studies, both with and without a comparison group. Studies with comparisons were classified into three possible variations: cluster randomized controlled trials, where randomization occurs at the group level; individual randomized controlled trials, where randomization occurs at the individual level; and non-randomized controlled trials, which involve no randomization. We also considered feasibility studies that proposed a descriptive framework. Some of these studies were supported by secondary data analysis and utilize existing datasets to extract valuable information to propose a descriptive framework. It is important to note that the objective of our SR is to provide an overview of the current state of the art and the level of maturity of individualized CT. Consequently, our criteria for study inclusion and exclusion were not restricted to particular research designs; in other words, we did not constrain the incorporation of studies with lower maturity, such as those lacking comparative analyses.

For each included controlled trial, we employed the SIGN methodology checklist, which presents a grading system ranging from 0 (not applicable) to 3 (well covered) for various items including participant assignment strategy, randomization, measurement types and validity, among others (see S1 Text: Appendix). This assessment resulted in a final grade that evaluates the extent to which the study was conducted to minimize bias, with grades of (++) indicating high quality, (+) indicating acceptable quality, (-) indicating low quality, and (- -) indicating unacceptable quality. The SIGN methodology proves to be a highly efficient rating system for assessing the quality of methodologies used in the included studies. Therefore, to compare the results of the SIGN analysis with the scores on our specifically designed scale, Table 4 displays a comparison between the SIGN risk of bias assessment, our customized evaluation of intervention quality, and the conclusions made by the authors.

**Table 3. Sample characteristics and characteristics of CT setup for the selected studies.**

| Study | CT features | | Within - CT measurements | | | CT effectiveness assessment | | Follow-up | Note |
|---|---|---|---|---|---|---|---|---|---|
| | Population | Content | Dosage | Location and training performance measures | Post and intra training subjective experience | Pre-post comparison | Cognitive Measurement (near, far effect (NFT) and everyday life transfer) | | Max = 11 |
| García-Rudolph and Gibert [12] | Acquired Brain Injury (ABI) and Traumatic Brain Injury (TBI) participants Age = [18–68] | Multi-domain PREVIRNEC system : rehabilitation tasks (attention, memory, executive functions, language) - 115 tasks | Duration : ND Frequency : ND Location : Home | No | No | Yes (**+1**) | NFT: Standardized NAB (28 tasks covering language, attention, memory, learning and executive functions) - source ND (**+1**) | No | 2 |
| Fermé et al. [97] (*Study protocol*) | NONE | Multi-domain 5 modules about knowledge (memory of stories, cancellation, questions of general knowledge, image pairs), comprehension (association, categorization), application (mazes, navigation); analysis (visual memory, word search); evaluation (comprehension of contexts..) - no task | Duration : ND Frequency : ND Location : Home | NA | NA | Yes | NFT: MoCA (short-term memory, executive functions, visuospatial abilities, language, attention, concentration, working memory, temporal and spatial orientation) | NA | NA |
| Xu et al. [98] | older adults free of mental disease/dementia/MCI Age = 63.7 ± 7 | Multi-domain Puzzle games (memory, attention, speed, visuo-spatial and executive functions) - 8 tasks | Duration : 2 weeks Frequency : 10 mn/per week (with 4 specific games) Location : Home (**+2**) | Yes (user adherence and preference) (**+1**) | Hand-made questionnaires - (elicited enjoyment, content and gaming mechanism preference, perceived difficulty and attention level) (**+1**) | Yes (**+1**) | NFT: MoCA (short-term memory, executive functions, visuospatial abilities, language, attention, concentration, working memory, temporal and spatial orientation) (**+2**) | No | 7 |
| Reidy et al. [99] | older adults free of mental disease/dementia/MCI Age = [60–100] | Multi-domain Virtual Reality based tasks: virtual supermarket (working memory) and multi-room museum (episodic memory) tasks - 2 tasks | Duration : 30 mn Frequency : 2 sessions of 15 minutes per day Location : laboratory (**+2**) | No | Standardized questionnaire - gaming experience questionnaire (immersion, engagement, flow) (**+2**) | Yes (**+1**) | NFT: Standardized NAB (spatial memory, perception, attention/orientation, memory, fluency, language) - source ND (**+1**) | No | 6 |

ND = Not Documented ; NA = Not Applicable; NAB = Neuropsychological Assessment Battery

**Table 3.** (Continued)

| Study | Population | CT features | | Within - CT measurements | | CT effectiveness assessment | | | Note |
|---|---|---|---|---|---|---|---|---|---|
| | | Content | Dosage | Location and training performance measures | Post and intra training subjective experience | Pre-post comparison | Cognitive Measurement (near, far effect (NFT) and everyday life transfer) | Follow-up | Max = 11 |
| Kitakoshi et al. (a) [100] | older adults Age = [70–90] | Specific domain Memory game - 1 task | Duration: 6 weeks Frequency: at least 5 min on participants behalf - 2 weeks per condition Location : Home (+2) | Yes (learning path and self-selected dosage) (+1) | Hand-made question-naires - (enjoyment, motivation, perceived difficulty) (+1) | No | No | No | 4 |
| Kitakoshi et al. (b) [101] | older adults Age = [70–90] | Specific domain Memory game - 1 task | Duration: 2 weeks Frequency: 10 min per day Location : Home (+2) | Yes (learning path) (+1) | Hand-made question-naires - (motivation and engagement) (+1) | No | No | No | 4 |
| Kitakoshi et al. (b) [101] | older adults Age = 79.2 ± ND | Specific domain Memory game - 1 task | Duration: 2 weeks Frequency: 10 min per day Location : Home (+2) | Yes (learning path and self-selected dosage) (+1) | Hand-made question-naires - (motivation and engagement) (+1) | No | No | No | 4 |
| Rathnayaka et al. [102] | adults with dementia Age = 76.4 ± ND | Multi-domain D-care (attention and concen-tration, executive functions, language and memory skills) - 4 tasks | Duration : 1 month Frequency: ND Location: ND | Yes (learning path) (+1) | No | No | No | No | 1 |
| Shen and Xu [103] | children Age = [10–11] | Multi-domain CogDaily (speed, memory, attention, flexibility, logic training) - 17 tasks | Duration: 2 weeks Frequency: about 15 min per day Location: Laboratory (+2) | No | No | Yes (+1) | NFT: Wechsler Memory Scale (processing speed and memory) (+2) | No | 5 |
| Sandeep et al. [104] (Data collection) | Young adults Age = 19.87 ± 2.32 | Specific domain N-back training ( "Tapback", "Recall" and "Recollect the study") - 3 tasks | Duration : 8-10 days Frequency: 16 to 20 ses-sions of 20 min with 2 sessions per day includ-ing a 10 min break between sequential sessions Location: Home | NA | NA | NA | NA | NA | NA |

ND = Not Documented ; NA = Not Applicable; NAB = Neuropsychological Assessment Battery

**Table 3.** (Continued)

| Study | CT features | | | Within - CT measurements | | CT effectiveness assessment | | | Note |
|---|---|---|---|---|---|---|---|---|---|
| | Population | Content | Dosage | Location and training performance measures | Post and intra training subjective experience | Pre-post comparison | Cognitive Measurement (near, far effect (NFT) and everyday life transfer) | Follow-up | |
| Sandeep et al. [104] (Data collection) | Young adults Age = 19.79 ± 1.87 | Specific domain N-back training ("Tapback", "Recall" and "Recollect the study") - 3 tasks | Duration : 8-10 days Frequency: 16 to 20 sessions of 20 min per day including a 10 min break between sequential sessions Location: Home | NA | NA | NA | NA | NA | Max = 11 |
| Wilms [105] | young adult with ABI Age = 49 | Specific domain VisATT (letter span and vision detection speed) - 1 task | Duration: 3 weeks Frequency: 30 min session per day Location: Home (+2) | Yes (learning path) (+1) | No | No | No | No | 3 |
| Solana et al. [106] | adults with cognitive decline (ABI) Age = [16–55] | Multi-domain Guttman Neuro Personal Trainer (GNPT, PREVIRNEC 2) (attention, memory, executive functions) - 95 tasks | Duration : 4 to 7 months Frequency: 2- 3 sessions of 1 hour per week with a number of total session of 60 Location: Home (+2) | Yes (learning path comparison) (+1) | No | Yes (+1) | NFT: Standardized NAB (attention, memory, executive functions) - source ND (+1) | No | 5 |
| Zini et al. [24] | young adults Age = 23.93 ± 2.29 | Specific domain MS-rehab: (alternating attention and working memory) - 1 task | Duration : ND Frequency: 20 types of exercise per session Location: Home (+1) | Yes (learning path) (+1) | No | Yes (+1) | NFT: PASAT (processing speed, working memory, sustained attention) (+2) | Yes - only for near effect (trained task) | 5 |
| Zedda et al. [107] | young adults Age = [28–45] | Specific domain cooking game (visual attention, working memory) - 1 task | Duration : 1 day Frequency: 45 minutes, Location : Laboratory, (+2) | No | Handmade questionnaires - user engagement (semi-structured interview about perceived differences between conditions, likeability, positive and negative aspects) (+1) | No | No | No | 3 |

ND = Not Documented ; NA = Not Applicable; NAB = Neuropsychological Assessment Battery

Table 3. (Continued)

| Study | Population | CT features | | Within - CT measurements | | CT effectiveness assessment | | | Note |
|---|---|---|---|---|---|---|---|---|---|
| | | Content | Dosage | Location and training performance measures | Post and intra training subjective experience | Pre-post comparison | Cognitive Measurement (near, far effect (NFT) and everyday life transfer) | Follow-up | Max = 11 |
| Eun et al. [108] | older adults Age = [60–90] | Multi-domain 4 modules (attention, logic, response time and memory) - 6 tasks | Duration: 8 weeks Frequency: no limit and ND Location : Laboratory (+1) | Yes (intra-training performance) (+1) | Hand-made questionnaires - satisfaction (engagement, fun, subjective performance) (+1) | Yes (+1) | No | No | 4 |
| Tsiakas et al. [109] | young (undergraduate and graduate students) Age = NA | Specific domain NIH Toolbox Cognition Battery (Working Memory test with socially assistive robots-based approaches) - 1 task | Duration: ND Frequency: Data collection - 20 minutes (including a post session user survey) Location: Laboratory | NA | NA | NA | NA | NA | NA |
| Book et al. [110] (*Study protocol*) | MCI Age = NA | Multi-domain MAKSCog (attention, executive function, perceptual-motor, executive functions, perceptual motor, language) - 10 tasks | Study protocol: Duration : 6 months and open phase in which participants can freely continue to use the CCTs Frequency : at least 30 min per day, 3 days a week Location : Home | No | Standardized questionnaire - User Engagement questionnaire (attractiveness, perspicuity, efficiency, dependability, stimulation and novelty of software), Hand-made questionnaire of usability | No | No | No | NA |
| Singh et al. [25] | older adults Age = 72.16 ± 5.5 | Multi-domain The Mind Frontiers cognitive training (Working memory updating, switching, dual N Back, TowerOfLondon, PipeMania, FigureWeights VisualSpatial) - 7 tasks | Duration : first period of 12 weeks (5 days out of 7) and second period of 6 weeks Frequency : Data collection - 45 minutes a day for phase 1, no limit for phase 2 Location: Home | Data collection - Yes (learning path) | Data collection - Hand-made questionnaires - (technical competence, subjective cognition, perceived benefits) | NA | NA | NA | NA |

ND = Not Documented ; NA = Not Applicable; NAB = Neuropsychological Assessment Battery

**Table 4. Risks of bias, proof level rating and authors conclusions.**

| Study | How well was the study done to minimize bias? | Is the over-all effect due to the study intervention? | Note on the custom scale | Summarise the authors' conclusions. Add any comments on your own assessment of the study, and the extent to which it answers your question and mention any areas of uncertainty raised above. |
|---|---|---|---|---|
| García-Rudolph and Gibert [12] | + | No | 2 | The authors introduced two methods for identifying neurorehabilitation (NRR) in patient samples. Both methods prioritize difficulty level and dosage to enhance rehabilitation effects. Clinical validation on a limited sample (n = 10) favors the sectorized annotated plan strategy for optimal difficulty targeting. |
| Fermé et al. [97] | - - | Not applicable | Not applicable | The authors proposed a framework to personalize the treatment of a cognitive rehabilitation tool. Their framework involves determining participants' cognitive profiles and employing a belief revision system for continuous cognitive level updates. This strategy aims to sustain an ideal difficulty level and motivation, yielding optimal rehabilitation outcomes. |
| Xu et al. [98] | + | Yes | 7 | The authors proposed an individualized serious game for cognitive training by employing a daily life recording strategy and intelligent techniques to incorporate visual lifelogs into training. The results indicate moderate effects on user adherence (significant difference in playing frequency on a Wilcoxon signed-rank test, p = 0.049, Hedge g = 0.39) and engagement (significant difference on paired t-test, t(25) = 3.410,p = 0.001) in favor of the personalized strategy. However, cognitive improvements were not observed (p = 0.691) (paired t-test, t(25) = -0.5, p = 0.691). While this marks a promising step towards optimal content individualization, the crossover design prevents efficacy assessment for cognitive improvement. |
| Reidy et al. [99] | + | Yes | 6 | The authors proposed a VR based CT and used intelligent strategies to automatically extract and classify affects from EMG data. Qualitative feedback analysis suggests that the individualized condition enhances feelings of competency and appropriate challenge. However, the study's limited sample size and crossover design preclude assessing cognitive improvement efficacy. |
| Kitakoshi et al. (a) [100] | + | No | 4 | The authors assessed the impact of a difficulty adjustment reinforcement learning algorithm (DA) and a break offering system (DABO). The study revealed higher enjoyment in the control condition and greater motivation in the DABO condition. Questionnaires indicated the DA algorithm offered suitable difficulty for most participants. Learning path analysis indicated appropriate difficulty levels in the DA condition. Nonetheless, the study's small sample size and crossover design hindered efficacy assessment for cognitive improvement. |
| Kitakoshi et al. (b) [101] | - | No | 4 | The authors proposed a personalized CT of memory through reinforcement learning. A preliminary study assessing the impact of 2 structures of the activity space (9 difficulty vs 13 difficulty levels) favored a lower number of difficulty levels. Interviews revealed inter-subject variability in optimal activity space perception, suggesting further investigation (study 2 below). |
| Kitakoshi et al. (b) [101] | - | No | 4 | The authors proposed individualized memory CT through reinforcement learning, comparing low-number (9) and high-number (13) difficulty level activity structures. Questionnaire analysis indicated that the high-number condition required less effort and allowed longer play sessions. Learning path and success rate analysis indicated high-number difficulty levels was better suited for difficulty adjustment algorithms. However, limited sample size and crossover design impeded assessing cognitive improvement efficacy. |

**Table 4.** (Continued)

| Study | How well was the study done to minimize bias? | Is the overall effect due to the study intervention? | Note on the custom scale | Summarise the authors' conclusions. Add any comments on your own assessment of the study, and the extent to which it answers your question and mention any areas of uncertainty raised above. |
|---|---|---|---|---|
| Rathnayaka et al. [102] | - | No | 1 | The authors proposed an individualized cognitive rehabilitation based on a reinforcement learning algorithm (Q-learning). The intervention group exhibited performance improvement across all proposed cognitive activities. Notably, the study lacks information about cognitive performance, subjective questionnaires, or inter-group comparisons. |
| Shen and Xu [103] | ++ | Yes | 5 | The authors proposed a recommendation algorithm for personalized cognitive training. Pre-test comparisons in cognitive performance showed no difference between groups (independent t-test,t(15) = 1.4, p>0.05 for processing speed and t(15) = -1.02, p = 0.32 for memory quotient). Post-test cognitive performance analysis revealed improvements only for the intervention group in processing speed (paired sampled t-test, t(15) = -2.62, p = 0.02) and in memory quotient (t(15) = -2.60, p = 0.02). |
| Sandeep et al. [104] | - - | Not applicable | Not applicable | The authors aimed to compare machine learning algorithms (Hidden Markov Models (HMM), Kalman filter (KF), and Long Short Term Memory (LSTM)) for predicting participant skill levels. Using data from a cognitive training intervention with the Recall game, history-driven HMM demonstrated better fit than HMM with a universal transition matrix (RMSE = 5.6%). Both HMM-based models effectively predicted skill levels. KF and LSTM estimated performance and skill levels but with weaker accuracy (RMSE = 18.83% and 9.34% respectively). |
| Sandeep et al. [104] | - | Not applicable | Not applicable | The authors aimed to compare machine learning algorithms (HMM, KF, and LSTM) for predicting participant skill levels. The dataset included diverse learning trajectories from a Recollect cognitive training intervention. HMM with a universal transition matrix displayed better fit (test RMSE = 12.54%) than history-driven HMM. Both HMM-based models effectively predicted skill levels. Study results differed from Study 1, revealing sensitivity to algorithm choice during initial data generation (difficulty adjustment procedure during initial intervention). KF and LSTM estimated performance and skill levels with less accuracy (RMSE = 31.52% and 18.77% respectively). |
| Wilms [105] | - | No | 3 | The author introduced a difficulty-adjusting reinforcement learning algorithm (actor-critic) for cognitive training. While the algorithm adapted difficulty levels, the study design precluded drawing conclusions about the approach's effectiveness. |
| Solana et al. [106] | + | Yes | 5 | The authors presented a clustering-recommendation strategy for individualized cognitive rehabilitation sequences. Comparisons of selected tasks and difficulty levels chosen by the intelligent strategy and the manual planning showed significant differences (p<0.001). No cognitive improvement disparities were observed between the two planning methods (p = 0.34). |

**Table 4.** (Continued)

| Study | How well was the study done to minimize bias? | Is the overall effect due to the study intervention? | Note on the custom scale | Summarise the authors' conclusions. Add any comments on your own assessment of the study, and the extent to which it answers your question and mention any areas of uncertainty raised above. |
|---|---|---|---|---|
| Zini et al. [24] | ++ | Yes | 5 | A reinforcement learning algorithm (SARSA) was proposed for individualized cognitive training. Results showed that participants started with homogeneous pre-test scores (2tailed Ttest, p = 0.42) and both groups improved after training (2-tailed paired T-test, p = 1.7 * $10^{-5}$ for group intervention and p = 0.02 for group control). Intervention participants using the RL algorithm showed greater cognitive improvement than control group (2-tailed T-test, p = 4 * $10^{-4}$). Learning trajectories indicated no significant success rate differences (2-tailed T-test, p = 0.56). On all trained tasks, the intervention group completed fewer activities on average than the control group. Follow-up evaluations demonstrated no between-group performance differences (task 1: p = 0.33, task 2: p = 0.06). An additional experiment with a modified RL algorithm (fine-tuned policy) showed no cognitive improvement differences but required fewer activities. |
| Zebda et al. [107] | - | No | 3 | The authors proposed individualized cognitive training via robot interactions using reinforcement learning (Q-learning). Multiple case studies highlighted participants' successful identification of the adaptive condition, with semi-structured interviews emphasizing participant enjoyment. |
| Eun et al. [108] | - - | No | 4 | This study introduced individualized cognitive training based on participant skill levels, utilizing a LSTM model for dynamic difficulty adjustment. The intervention group exhibited improved quality of life, certain geriatric depression test components, and mini-mental status examination results. Pre-post cognitive performance comparison showed significant improvement in all cognitive activities (except one) (repeated measure ANOVA, t = 2.76 p = 0.006 for memory training, t = 5.94, p = 0.00 for vision adaptation, t = 10.4, p = 0,000 for icon training, t = 5.423 p = 0.000 for graph training). The study design did not allow for separating the personalized procedure's impact from the training program itself. |
| Tsiakas et al. [109] | - - | Not applicable | Not applicable | The authors introduced socially assistive robots for cognitive training (CT), which tailor learning by monitoring task engagement and performance. Their approach involves modeling artificial participants, training reinforcement learning (RL) models, and assessing them in a virtual environment. Results indicate RL models effectively generate distinct policies for various user profiles. |
| Book et al. [110] | - - | Not applicable | Not applicable | This study suggests an individualized cognitive training design based on performance prediction through logistic regression. However, no data is provided to support the proposal (study protocol). |
| Singh et al. [25] | - - | Not applicable | Not applicable | The authors presented data augmentation techniques and deep-learning strategies (CNN, LSTM, CNN-LSTM) for predicting adherence to cognitive training. Model fitting showed successful training and prediction on the dataset, with approximately 75% accuracy, AUC, and F-score. |

**Table 5. Risks of bias, proof level rating and authors conclusions (non comparative and feasibility studies are not included).**

| Study | Appropriate and clearly focused question | Randomized assignment to treatment group | Adequate conceal-ment method | Double blind allocation | Homogeneity between groups | Only dif-ference is treatment | Standard, valid and reliable measures | Percentage of dropouts | Intention to treat analysis | Valid multi-sites comparison | Grade |
|---|---|---|---|---|---|---|---|---|---|---|---|
| García-Rudolph and Gibert [12] | 3 | 1 | 2 | 1 | 3 | 3 | 3 | 2 | 3 | 0 | + |
| Xu et al. [98] | 3 | 3 | 3 | 1 | 1 | 3 | 2 | 2 | 3 | 0 | + |
| Reidy et al. [99] | 2 | 2 | 3 | 1 | 1 | 3 | 2 | 2 | 3 | 0 | + |
| Kitakoshi et al. (a) [101] | 2 | 2 | 3 | 1 | 1 | 3 | 2 | 2 | 3 | 0 | - |
| Kitakoshi et al. (b) [100] | 2 | 2 | 1 | 1 | 1 | 3 | 2 | 2 | 3 | 0 | + |
| Shen and Xu [103] | 3 | 3 | 3 | 3 | 3 | 2 | 2 | 2 | 3 | 0 | ++ |
| Solana et al. [106] | 3 | 1 | 3 | 1 | 3 | 3 | 3 | 2 | 3 | 0 | + |
| Zini et al. [24] | 3 | 3 | 3 | 3 | 3 | 3 | 2 | 2 | 3 | 0 | ++ |
| Zebda et al. [107] | 3 | 2 | 2 | 1 | 1 | 3 | 1 | 2 | 3 | 0 | - |

## 3 Results

### 3.1 Descriptive results

The systematic review processed seventeen papers including nineteen studies (12 in journals and 7 in proceedings articles) that have been published from 2011 to 2022 (Table 2). Almost 70% (n = 13) of included papers were published during the last three years demonstrating the relatively low maturity of the field. Six studies are non-empirical, presenting either a study protocol (n = 2, [97,110]) or a feasibility study (formal validation) evaluating new methods on existing datasets (n = 4, [25,104,109]). According to the SIGN methodology for study design [112], eleven used a controlled trial (n = 8 non-randomized controlled trial and n = 3 Individual randomized controlled trial) and two proposed either a case study [105] or a non-comparative study [108]. All the controlled trials included an active control group, either using a between-subject or within-subject design. During the intervention, the participants engaged in the same training as the intervention group, but without any adaptive procedure. It is noteworthy that none of the studies included passive control conditions where no intervention was implemented. Among the total of nineteen studies, approximately 70% (n = 13) aimed to assess CT with non-clinical samples, while the remaining 30% (n = 6) had a rehabilitative objective and investigated clinical samples. When assessing the research conducted on actual populations, the average sample size was 85, and the median sample size was 20. Nevertheless, within the three studies with the largest participant pools, [12] presented two cohorts consisting of n = 123 and n = 327 individuals. Notably, the individualized CT was exclusively examined in a subgroup of n = 10 participants within the treatment condition. Factoring in this information, the mean sample size adjusted to 60, with the median sample size reduced to 10.

### 3.2 Q1 & Q2. What Type of AI Techniques have been used in the field of computerized CT? What are the Subject/Domains of CT for which adaptive techniques have been designed?

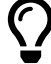

**Main results:**

- A quarter of the included strategies employed a macro-adaptive approach, all of which were for multi-domain cognitive training.
- The majority of papers presented micro-adaptive procedures, either for predicting the next optimal learning activity directly or for extracting patterns to inform optimal pedagogical decisions.
- Half of the micro-adaptive procedures targeted a single, cross-cutting cognitive function, while the remaining half employed a multi-domain approach.

Among the nineteen studies examined, only 26% of them (n = 5) put forth the utilization of a macro-adaptive procedure to customize the intervention. For example, [12] aimed to improve the understanding of optimal learning objectives. Specifically they used visual annotated plans and decision trees techniques to identify the range of difficulty known as the "neurorehabilitation range" (NRR). Other approaches suggested tailoring the entire curriculum in advance through a recommendation system that leveraged participant similarities [98] or by employing clustering techniques to identify cognitive profiles [103]. Another proposal involved directly customizing the visual content of cognitive activities by utilizing automatic extraction of relevant images from daily visual logs [98].

The 74% (n = 14) of remaining papers used a micro-adaptive approach with different strategies. These studies can be broadly categorized into two groups. The first category encompasses eight studies that primarily concentrated on directly predicting the next optimal activity by tailoring the difficulty level or the game content. Reinforcement learning methods were commonly used, with three different algorithms employed: Q-learning (n = 4), Bucket brigade (n = 3), and Actor-critic (n = 1). Additionally, [108] proposed a method utilizing deep learning, particularly Long Short-Term Memory (LSTM). The second category comprises six intelligent methods designed to extract valuable information from collected data, facilitating the generation of optimal pedagogical decisions. In all studies within this category, the choice of the next activity is based on expert hand-designed heuristics or algorithms. It is worth noting that among the studies in this category, four of them are feasibility studies without evaluating a real population. One approach aims to predict participants' performance on the next activities based on their previous trajectory. For this purpose, [104] proposed Bayesian techniques such as hidden Markov models and Kalman filters, as well as deep learning utilizing LSTM. Another strategy involved employing machine learning techniques, specifically logistic regression, to predict participant performance [110]. In addition, [25] utilized deep learning algorithms (LSTM and CNN) directly to infer the probability of dropout in the next activities. Finally, [99] suggested using deep learning techniques to extract useful information from EMG data.

As demonstrated by Fig 3, half of the studies (n = 10) adopted a multiple cognitive domain approach for designing the CT. Among the nine studies focusing on single domain; the targeted functions were attention [105] or working memory [24,104], i.e., cognitive functions that are seen as cross-cutting to many other cognitive functions or activities, and are therefore expected in a CT to improve a large number of cognitive domains. It is noteworthy that studies using specific domain training used mostly a micro-adaptive approach with RL techniques. Consequently, these observations indicate that micro-adaptive strategies are preferred for

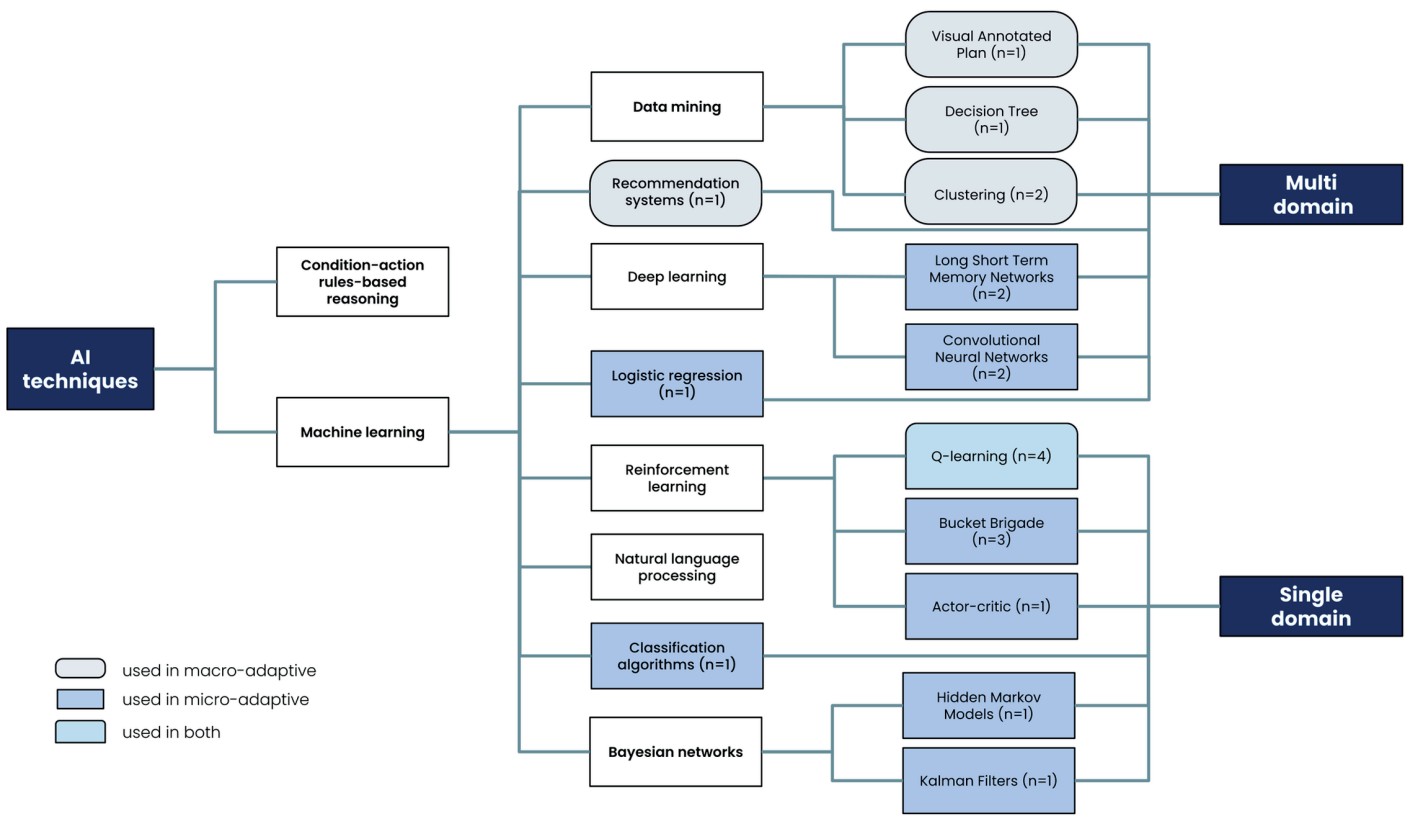

**Fig 3. Distribution of AI techniques depending on type of CT studied (multi or single domain).**

https://doi.org/10.1371/journal.pcbi.0316860.g003

the single cognitive domain CTs while macro-adaptive strategies are preferred for CTs with multiple cognitive domains.

## 3.3 Q3. What populations are targeted and what are the characteristics of the CT design?

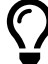

**Main results:**

- The majority of studies involved non-clinical adult populations.
- Experimental designs varied widely, with many conducted remotely, and no clear patterns emerged regarding intervention duration, frequency, or assessment strategies.

Among the thirteen studies with non clinical samples, twelve of them included adults (n = 7 with older adults and n = 5 with young adults) for whom specific domain CTs (n = 7) were performed rather than multiple-domain ones (n = 4). The only study including children performed a multiple-domains CT (Table 3). Among the six studies with a rehabilitative purpose, three of them included young adults with acquired brain injury (ABI) or traumatic brain injuries (TBI), two of them involved older adults with dementia or mild cognitive impairment (MCI). For these two types of clinical samples, the multiple-cognitive domains approach has been widely used (80%, n = 4). The remaining study [97] proposed a general framework

that is agnostic to a specific population. Taken together, the selected studies mirrors well the two distinct literature, where CTs are often single-domain by targeting a cross-cutting function (attention, working memory) (e.g., [23], and cognitive rehabilitation programs are rather multiple-domains, as this intervention design has been shown to be more clinically effective than single domain interventions (e.g., [113,114]). In relation to the CT settings outlined in Table 3, the majority of interventions were carried out remotely at participants' homes (n = 9), while others took place in laboratory settings (n = 4), and information was not provided for (n = 2) cases. The time duration of the CT varied significantly, ranging from lengthy periods of seven months to brief sessions of only thirty minutes. However, the most commonly reported duration was two weeks (n = 4), and in some studies, information regarding the duration was not available (n = 4). Among the studies that documented the CT dose (n = 10), there was substantial variation observed, with session duration ranging from five minutes to one hour per day. Cumulative sessions encompassed a wide range, from 30 minutes (n = 1) to over 600 minutes (n = 3), often with intermediate duration averaging around 140 to 215 minutes or 2 hours and 30 minutes to 3 hours and 30 minutes (n = 5). A total of 48% of the studies (n = 9) intended to document the training effect throughout the experiment using objective measures of performance or participants' subjective experiences related to the intervention. Regarding the assessment of participants' subjective experiences (n = 8), the majority of studies (n = 6) relied on manual evaluations (non standardized measurements). The subjective evaluations were related to several dimensions such as engagement, game preferences, motivation or perceived difficulty. In six studies, pre- and post-intervention comparisons of cognitive performance were conducted.

### 3.4 Q4. How effective are individualized techniques in empirical CT studies? What effects are reported (NFT and everyday life transfer effects)? Are the effects dependent on the CT design (content, dose, location) and the targeted sample?

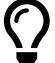

**Main results:**

- In one study, no distinctions were found between groups in pre-post assessments, highlighting that an automated individualized procedure exhibited equivalent efficacy to a manual approach.
- Two studies exhibited more substantial cognitive enhancements in post-test measurements, specifically in near-transfer measures.
- Several studies utilized non-comparative or cross-over designs, making it challenging to differentiate the training's impact in pre-post assessments.
- Five studies exhibited varying learning trajectories through intra-training measures, while six showed subjective differences in motivation, engagement, and play frequency between individualized and control groups.

Various dimensions were considered to illustrate the effectiveness of empirical CT studies, as shown in Fig 4. Firstly, out of the seven studies that aimed to evaluate the progression of cognitive performance using pre-post assessments, three interventions [98,99,108] employed either a crossover or a non comparative design, making it challenging to distinguish the impact of the control procedure from the individualized approach on cognitive performance. In the other hand, no significant differences in cognitive enhancement were found between the intervention and control groups in [106], indicating that the automated procedure's effectiveness matches that of the manual approach across a comprehensive neuropsychological assessment battery. Furthermore, by ensuring group homogeneity during the pre-test, both [103] and [24] demonstrated that the personalized approach resulted in

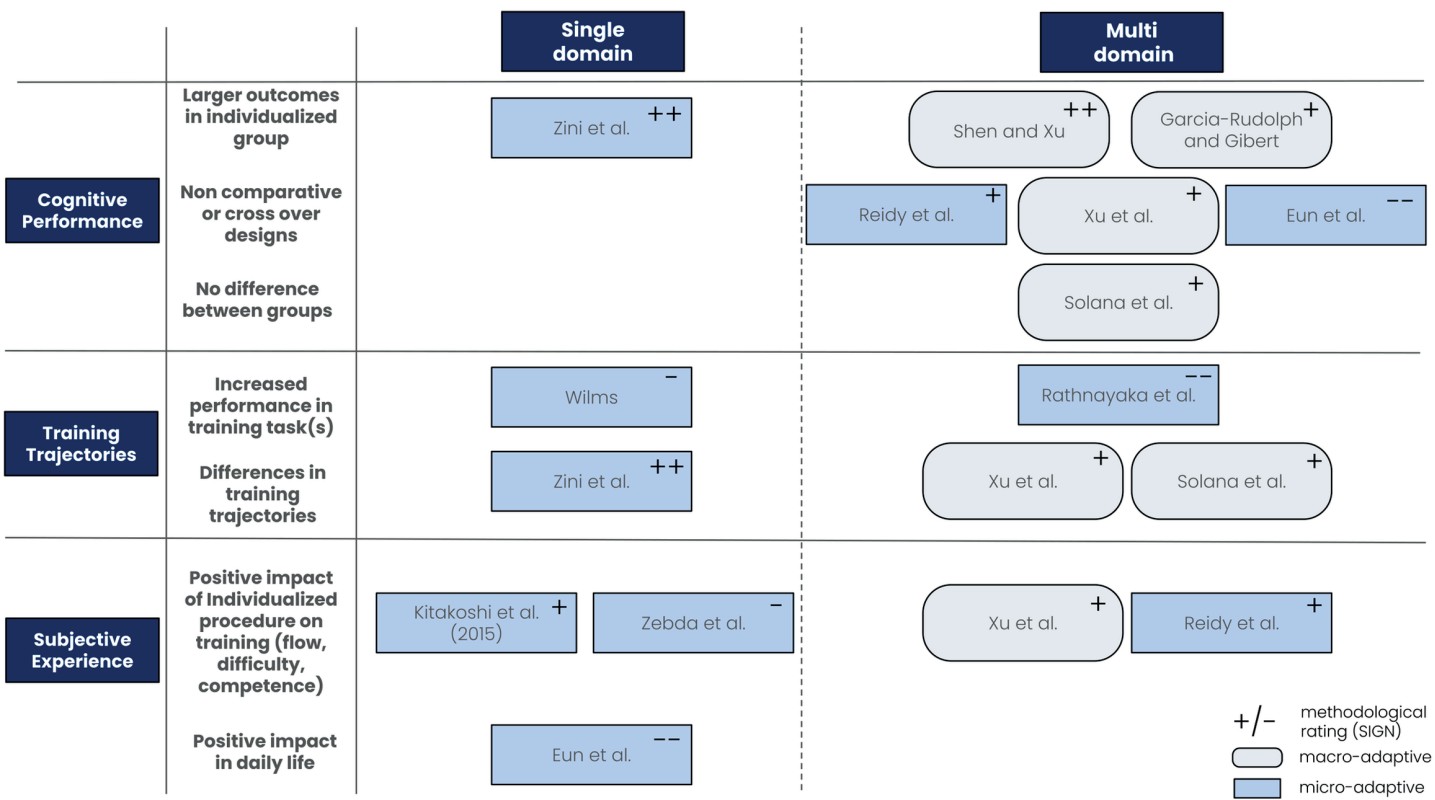

**Fig 4. Comparison of studies across single and multi-domain interventions in Cognitive Performance, Training Trajectories, and Subjective Experience.** Studies are classified by adaptive strategies (macro-adaptive in gray, micro-adaptive in blue) and rated for methodological quality using the SIGN scale (+/- symbols). "+" indicates higher methodological quality, while "-" indicates lower quality.

https://doi.org/10.1371/journal.pcbi.0316860.g004

more pronounced cognitive changes concerning measures of near transfer. [24] also conducted a follow-up evaluation on the trained task but were not able to see any difference in performance between groups. Furthermore, [12] observed a significant improvement of performance for a small subset of participants treated with an optimal difficulty level. It is noteworthy that the 3 interventions showing significant changes were all using different CT programs (multi and single domain, different dosage, laboratory and at home based, population of healthy young adults, children and ABI patients…) and different cognitive evaluations.

Another aspect leveraged to assess the impact of the proposed intervention was to observe quantitative intra-training measures. First, two studies [102,105], presented an increase in the performance on the trained task as a proxy for cognitive evolution. Then other authors showed how the individualized procedures affected the learning path proposed. [100,101, 106] performed a comparative analysis of the learning trajectories of the non-adaptive control group and the treatment group, revealing notable differences in the patterns of learning. Moreover, analysis of quantitative intra-training observations revealed differences in the schedule of activity proposed: [98] showed a significant increase in the self-management of playing frequency with the individualized game compared to the non personalized but no significant difference in intensity (average sessions length). Additionally, [24] demonstrated a significant disparity in the number of episodes played, indicating that the

individualized procedure facilitated greater cognitive improvement in a shorter period of time.

To gain insights into the impact of the intervention, subjective measures were also employed. Firstly, [99] utilized the Game Experience Questionnaire [115] demonstrating that the individualized procedure positively influenced the participants' sense of competence. The intervention also led to a better-suited level of difficulty, as evidenced by an increase in flow and a decrease in the feeling of challenge. Then [108] showcased various positive impacts of individualized CT. Participants reported an improvement in subjective health condition and overall quality of life. Moreover, there was a reduction in certain items of the Geriatric Depression Scale Short Form (GDSSF-K, [116]) and a positive change in the Mini-Mental State Examination (MMSE, [117]). It is worth noting that these results were not compared with an active control trial. Additionally, three other studies utilized custom-made questionnaires to assess subjective performance. [98] found that participants using the individualized procedure experienced higher enjoyment, which was further substantiated by qualitative feedback obtained through interviews. [100] revealed that the individualized intervention fostered greater motivation to use the system and maintained a suitable difficulty level. Furthermore, [107] interviews indicated that the individualized procedure was perceived as more stimulating and engaging.

### 3.5 Q5. What Type of Validation have been conducted for these new generations of computerized CT?

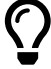

**Main results:**

- In accordance with SIGN ratings, two studies were rated as (++), five as (+), four as (-), and eight as (- -).
- The mean score on the customized scale was 4.1 out of 11.
- Increased scores on the customized scale corresponded to higher ratings on the SIGN rating scale.

Based on the SIGN rating, it was found that out of the nineteen studies examined, only two received the highest score (++) [24,103]. Five studies received an acceptable score (+). Within this group, two randomized controlled trials [12,106] focused on clinical samples, and the reason for not receiving a (++) grade was due to issues related to participant randomization and blinding. The remaining three studies [98–100] received a (+) grade primarily because of their implementation of a crossover design.

Among the nineteen studies, four studies received a (-) score. Two of these studies [101] utilized a crossover design but lacked important information in their reports (see S1 Text: Appendix), while the other two studies [105,107] did not include any control group. Additionally, eight studies received a (- -) score. Six of these studies were either proposing a study protocol or conducting a feasibility study. The remaining two interventions [102,108] were assigned a (- -) score due to the absence of a control group and a lack of important information (see S1 Text: Appendix).

Studies that obtained a (++) score achieved an average score of 5 on the customized scale, while studies with a (+) score had an average score of 4.8. For studies receiving a (-) score, the average score was 3.5, and for studies with a (- -) score, the average score was 2.5 (whenever applicable). These findings emphasize a noticeable correlation between risk evaluations and the number of standards fulfilled in CT research. Furthermore, the results demonstrate that the majority of studies (n = 12) did not meet the acceptable criterion of the SIGN

methodology (+), and none of the studies fulfilled all the standards outlined by the customized scale. The average score across all studies was 4.1.

Results from the SIGN ratings strategy were included in Fig 4 to compare study outcomes with their corresponding methodological quality.

## 4 Discussion

This SR explored the wide array of AI techniques employed to enhance individualized CT. To begin with, the deployment of macro-adaptive strategies, which may draw from participant resemblances or the formation of cognitive profiles, facilitates the utilization of existing knowledge in the development of individualized schedules for cognitive tasks. These approaches prove particularly valuable when implementing multi-domain CT, especially when multiple cognitive processes are engaged across various activities. As evidenced in various cognitive rehabilitation studies [12,97], health professionals often face challenges in selecting the most suitable sequence of activities. Consequently, employing macro-adaptive strategies that can leverage data from large cohorts presents a promising avenue for developing effective interventions. Furthermore, as suggested by [25], these methods offer insights into the mechanisms underlying improvements and adherence to the interventions. Nevertheless, macro-adaptive procedures, by their inherent nature, do not entail direct adaptation of the content and difficulty levels of individual tasks. Instead, they yield more intricate outcomes that require comprehensive analysis. Consequently, given the still exploratory state of the field, the majority of studies reviewed here has focused on tailoring single or a few training tasks using a micro-adaptive approach. Micro-adaptive procedures propose to use previous interaction with the user to personalize the learning trajectory. Most modern AI techniques leverage collected data from the training path and thus fit particularly well with the CT paradigm where many short episodes are played. While this task may appear less challenging than planning a complete curriculum in advance, it requires data-efficient strategies to identify and suggest activities with appropriate dynamics for tailoring the path to each participant's needs. As a result, many studies employing deep learning or machine learning techniques are still undergoing formal validation and are currently in the feasibility study stage, being tested solely on previous data and not yet evaluated on real participants [25]. Reinforcement learning paradigm, where the artificial teacher, or system, proposes activities based on the participant's previous interactions looks like a particularly good fit for that purpose but also has its limitations: to enhance data efficiency, most strategies rely on tabular approaches, which in turn restrict the number of parameters available for adaptation. Moreover, for several studies of this SR, a two-stage time consuming strategy is commonly employed where a first teacher policy is being trained on a group of participants and is then fine-tuned for each participant (e.g., [24]). Finally, a third family of strategies based on recommendation algorithms show promise but also require sufficient pre-collected data to achieve efficiency in personalizing the training experience.

Additionally, it is noteworthy to observe that most micro-adaptive strategies propose a personalization based on the difficulty of the cognitive tasks. As proposed by [77,78,118], the key idea is to propose an optimal difficulty in order to foster training gains and motivation and is tightly connected with the optimal cognitive challenge [68]. For that purpose, while many studies primarily focus on choosing the correct parameter set, certain approaches suggest modifying the content according to participants' visual cues. This alternative approach to customizing training harmonizes effectively with Mayer's Cognitive Theory of Multimedia Learning and his personalization principle [119]. Along this line of customization, the adaptation of interactions through assistive robots [107], chatbots [100] or virtual reality [99] is likely to be

another key factor for participant engagement. In this direction, it is conceivable that recent advancements with large language models will enable better dialogic adaptation, potentially impacting motivation and engagement [120]. However the issue of reproducibility becomes increasingly significant when incorporating complex data-driven strategies. Ensuring the transferability of models and reproducibility of experiments raises a challenge in the absence of provided code or dataset accessibility across the included papers. This lack of transparency is of growing criticality for research reproducibility. Consequently, the field of AI frequently encounters a black box scenario, which hampers reproducibility efforts. In the context of CT and its human stakes, it is important to understand the methods used to individualize the training path for each trainer, and if these are not sufficiently transparent, they must at least be traced or documented as predictors of targeted intra-training mechanics.

The findings of the present SR highlighted the current state of individualized CT as a field with relatively low maturity. Following the recommendation put forth by [111], there exists an urgent requirement to clarify the objectives of each study within the CT community. To achieve this, [111] proposed a distinction among several categories: feasibility, aimed at "testing the viability of a particular paradigm or project"; mechanistic, focused on "identifying the mechanism(s) of action of a behavioral intervention for cognitive enhancement"; efficacy, with the goal of "validating an intervention as the primary cause of cognitive improvements beyond any placebo or expectation-related effects"; and effectiveness, concerned with evaluating whether a given intervention "achieves the desired and predicted positive impact, often involving real-world outcomes". Notably, none of the studies encompassed in this review employed such terminology, yet it becomes obvious that the majority of interventions are currently positioned at the feasibility or mechanistic study stages. This observation is reinforced by the predominant focus of these interventions on non-clinical populations, specifically targeting young adults. Moreover, it is essential to note that very few studies adhered to the gold standard of Randomized Controlled Trials (RCTs). While RCTs have certain limitations, such as the need for stable, long-term interventions spanning several years to establish robust scientific evidence, they remain a crucial benchmark for evaluating interventions [121].

Specifically in the context of individualized interventions, mere observation of favorable and definitive outcomes arising from an individualization algorithm in the context of pre-post training effects is insufficient. What is imperative is the ability to elucidate its impact on the active cognitive mechanisms underpinning the training process, and subsequently, to establish a coherent connection between these mechanisms and the resultant effectiveness. A deep understanding of the causal relationships existing between the behaviors governing individualization and the intricate mechanics of training, as well as their collective impact on training effectiveness, stands as an essential foundation for the advancement of these emerging computerized cognitive therapies. To attain this level of understanding, the incorporation of judicious supplementary evaluations holds utmost significance. These assessments should aim to gain a comprehensive understanding of algorithm behavior, allowing researchers to gauge the effectiveness and adaptability of the individualized interventions. Moreover, the integration of subjective questionnaires can help evaluate participants' motivation and engagement levels, providing valuable insights into their experiences and receptiveness to the intervention. Such subjective metrics also possess the potential to shed light on how the customization of training can serve as an efficient mechanism for enhancing participants commitment to the program, thereby potentially mitigating the unfortunate phenomenon of attrition, which regrettably tends to manifest, particularly among older adults or clinical cohorts, who nonetheless manifest a demonstrable necessity for the training regimen [1].

The field of individualized CT, as depicted in this SR, mirrors the broader literature on CT, which is characterized by methodological and empirical weaknesses in assessing intervention effectiveness leading to controversy among experts [122]. This review highlights significant heterogeneity in methods, cognitive domains, dosage, and study populations, aligning with findings from other studies appealing for a greater compliance with more rigorous methodological standards. Unlike prior meta-analyses that presented mixed results regarding the dose-dependency of training effects in CT (e.g at least 10 sessions for [34] or 3 or fewer sessions in [19], our study does not provide evidence supporting a particular direction. Additionally, while previous studies (e.g., [114]) have indicated that multi-component training may exhibit greater efficacy compared to single-component training, half of the investigations included in this review primarily concentrate on attention or working memory functions. The emphasis on these functions is justified by their crucial cross-cutting role in everyday activities and their vulnerability to impairment in various cognitive pathologies [1]. Lastly, as emphasized in several systematic reviews [19,23], and the reanalysis of 2018 [123], the definition of a suitable cognitive battery that assesses NFT and ecological transfer significantly influences the measured outcomes and the conclusions drawn. Notably, improvements in certain cognitive domains might not manifest when assessed using different cognitive tasks (see [123]). Furthermore, it is essential to keep in mind that enhancements observed in a specific cognitive domain do not necessarily guarantee true transfer, as evidenced by the case of verbal memory training and its effects on neuropsychological tests [23]. The studies included in this review underscore the substantial diversity in assessment methods and the limited availability of approaches to evaluate broader ecological transfer.

## 5 Conclusion

The present systematic review puts forth a range of potential methodologies to better address interindividual differences and offers captivating prospects for the future development of the field. The hypothesis of heightened engagement and motivation found support in the limited number of studies that investigated this aspect. Further investigations are necessary to validate whether AI strategies can truly empower each participant's cognitive potential, and then ensure CT benefits for all. Although additional research endeavors adhering rigorously to methodological standards are still required, the first results appear promising. In line with this drive for progress, a notable observation emerged during the course of this systematic review: the number of included papers nearly doubled, particularly in the year 2022. This indicates a growing interest in individualized cognitive training and underscores the optimistic outlook for the field's future.

## Supporting information

**S1 Text: Appendix.**
(PDF)

**S2 Text: PRISMA checklist.** A Prisma-p checklist has been completed in addition to the PROSPERO registration.
(PDF) .

## Acknowledgments

First, we express our gratitude to onepoint and the R&D team, especially Erwan Le Bronec, which supported us to carry out this work.

## Author contributions

**Conceptualization:** Maxime Adolphe, Masataka Sawayama, Hélène Sauzéon.

**Data curation:** Maxime Adolphe, Marion Pech, Masataka Sawayama.

**Formal analysis:** Maxime Adolphe, Marion Pech, Masataka Sawayama, Hélène Sauzéon.

**Funding acquisition:** Denis Maurel, Alexandra Delmas, Pierre-Yves Oudeyer, Hélène Sauzéon.

**Investigation:** Maxime Adolphe.

**Methodology:** Maxime Adolphe, Masataka Sawayama, Denis Maurel, Alexandra Delmas, Hélène Sauzéon.

**Project administration:** Denis Maurel, Alexandra Delmas, Pierre-Yves Oudeyer, Hélène Sauzéon.

**Resources:** Hélène Sauzéon.

**Software:** Hélène Sauzéon.

**Supervision:** Denis Maurel, Alexandra Delmas, Pierre-Yves Oudeyer, Hélène Sauzéon.

**Validation:** Hélène Sauzéon.

**Visualization:** Maxime Adolphe.

**Writing – original draft:** Maxime Adolphe, Marion Pech, Pierre-Yves Oudeyer, Hélène Sauzéon.

**Writing – review & editing:** Maxime Adolphe, Marion Pech, Masataka Sawayama, Denis Maurel, Alexandra Delmas, Pierre-Yves Oudeyer, Hélène Sauzéon.

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
