## [Decision Letter · Decision Letter 0]

10 Sep 2024

PONE-D-23-33514Exploring the potential of artificial intelligence in individualized cognitive training: A systematic reviewPLOS ONE

Dear Dr. Adolphe,

Thank you for submitting your manuscript to PLOS ONE. After careful consideration, we feel that it has merit but does not fully meet PLOS ONE’s publication criteria as it currently stands. Therefore, we invite you to submit a revised version of the manuscript that addresses the points raised during the review process.

**ACADEMIC EDITOR: Please insert comments here and delete this placeholder text when finished.** Be sure to:

reshape some paragraphs according to Reviewer_2's and Reviewer_4's recommendations.move the description of AI earlier in the paper. check all tables's contents as suggested by reviewers. 

We look forward to receiving your revised manuscript.

Kind regards,

Alessandro Bruno, Ph.D.

Academic Editor

PLOS ONE

Journal Requirements:

Additional Editor Comments:

Dear Authors,

I appreciate the changes you've made to the manuscript, which is far better.

Therefore, considering the reviewers' report, I recommend it for a Minor Revision round.

Once you've worked out the minor issues below, the paper will be ready for acceptance.

Sincerely

A.B.

Please look into the following remarks by Reviewer 2:

-Perhaps the research questions could be reduced.

-I think the use of AI should be mentioned much earlier in the Introduction. As it currently reads, it is not mentioned until page 6.

-Table 2 that reviews the papers should include age ranges. I think it may also be better to focus just on adult population studies rather than child, adult and older adults.

Please address the comments by Reviewer 4

-The flowchart (fig.1) is unclear in some parts, e.g. in the sections ‘studies excluded’ and ‘studies sought for retrieval’ it would be helpful to specify the reasons. In the section ‘studies included’, 17 articles are reported, but it is not made clear that these are 19 studies. Please review and add these.

-In the paragraph Sorting keys of AI techniques for content adapting to learner’s capabilities the authors report that AI techniques can be classified into four main families, yet they only list three of them. Please make this information consistent.

-Please check that all acronyms in the paper are spelled out in full at least once, e.g. in the paragraph Evaluation of AI techniques the acronym NFT is not specified.

-The table 5 only shows 9 out of 19 studies, no reason is specified. Please clarify this point.

Reviewers' comments:

Reviewer's Responses to Questions

**Comments to the Author**

1. Is the manuscript technically sound, and do the data support the conclusions?

Reviewer #1: Partly

Reviewer #2: Partly

Reviewer #3: Partly

Reviewer #4: Yes

2. Has the statistical analysis been performed appropriately and rigorously? 

Reviewer #1: N/A

Reviewer #2: N/A

Reviewer #3: N/A

Reviewer #4: N/A

3. Have the authors made all data underlying the findings in their manuscript fully available?

Reviewer #1: Yes

Reviewer #2: Yes

Reviewer #3: Yes

Reviewer #4: Yes

4. Is the manuscript presented in an intelligible fashion and written in standard English?

Reviewer #1: Yes

Reviewer #2: Yes

Reviewer #3: Yes

Reviewer #4: Yes

5. Review Comments to the Author

Reviewer #1: Problems identified with the Paper

1. The abstract underscores the pressing issue of responder heterogeneity, revealing that individuals exhibit diverse responses to cognitive training interventions. This stark diversity in response necessitates the immediate adoption of individualized curriculum designs, moving away from generic, one-size-fits-all approaches.

2. The abstract highlights methodological differences across the selected studies, suggesting research design, implementation, and measurement variability. Weaknesses such as the absence of control groups and small sample sizes are noted, which can undermine the reliability and validity of study findings.

3. Despite the promising results observed in some studies, there needs to be a recognized gap in fully understanding and empirically supporting individualized techniques in cognitive training. This indicates a need for further research to explore the effectiveness, mechanisms, and long-term impacts of AI-based individualized approaches.

4. The abstract mentions areas for improvement in the design of the reviewed studies, such as the need for more control groups and small sample sizes. These methodological limitations can limit the generalizability and robustness of study findings, posing challenges to drawing firm conclusions about the efficacy of AI-based individualized cognitive training approaches.

5. While the abstract discusses analyzing intra-training performance as an outcome measure, it also hints the crucial need for more sophisticated and reliable assessment methods. These methods are essential to accurately track and assess individuals' performance and progress during cognitive training sessions, capturing even the most subtle changes in cognitive functioning over time.

6. The study mentions three main research questions (Q1-Q5), but the specific formulation of these questions needs to be clearly articulated. This lack of clarity makes it difficult to understand the precise focus of each research question and how they relate to the overall objectives of the study.

7. The study primarily focuses on describing existing individualization strategies in computerized cognitive training (CT) tools, understanding researchers' motivations for employing these strategies, and evaluating the effectiveness of included studies. However, it needs to provide a comprehensive overview of the broader context or theoretical frameworks guiding the research. This limited scope may lead to gaps in understanding and interpretation.

8. The methodology section needs more detail regarding the procedures employed in conducting the systematic review. There is no mention of search strategies, inclusion/exclusion criteria for selecting studies, data extraction methods, or quality assessment criteria. This lack of transparency raises concerns about the rigor and reproducibility of the review process.

9. While the study aims to evaluate the effectiveness of included studies in light of their design and statistical power, it needs to provide a clear framework for assessing study quality. Specific criteria or tools used to evaluate the methodological rigor of individual studies must be mentioned, which may compromise the reliability and validity of the findings.

10. The discussion attempts to navigate the complexities of macro-adaptive and micro-adaptive strategies in individualized cognitive training. These concepts involve intricate data analysis and adaptation processes, which may be challenging for readers unfamiliar with the field to grasp fully. Simplifying these concepts without oversimplifying their significance poses a challenge.

11. The discussion addresses the state of individualized cognitive training as a field with relatively low maturity. However, the findings may need more generalizability due to the predominance of studies targeting non-clinical populations, specifically young adults. This limitation restricts the applicability of the research findings to broader demographic groups or clinical settings.

12. The discussion acknowledges methodological and empirical weaknesses within cognitive training research, including heterogeneity in methods, cognitive domains, dosage, and study populations. Addressing these weaknesses requires a concerted effort to enhance the rigor and reliability of future research endeavors, which may pose significant challenges given the current landscape of the field.

13. The discussion highlights challenges in assessing intervention effectiveness in individualized cognitive training, particularly regarding the causal relationships between behaviors governing individualization and training outcomes. Establishing these relationships requires sophisticated evaluation methods and a deep understanding of underlying cognitive mechanisms, which may be difficult to achieve due to the complexity of cognitive processes and individual variability.

14. The discussion underscores the need for greater compliance with rigorous methodological standards in cognitive training research, including the definition of suitable cognitive batteries and assessment methods. Achieving consensus on standardized evaluation protocols and methodologies presents a significant challenge due to the diverse nature of cognitive training interventions and outcome measures.

Reviewer #2: Thank you for the opportunity to review this paper. It covers a systematic review of the use of AI in cognitive training. Briefly I have a few recommendations that may improve this paper for a future submission.

(1) Although I appreciate the amount of work that has been achieved, I found the paper to be far too long. Perhaps the authors could look at reducing. I found it hard to follow and overly complicated. Perhaps the research questions could be reduced.

(2) I think the use of AI should be mentioned much earlier in the Introduction. As it currently reads, it is not mentioned until page 6.

(3) Table 2 that reviews the papers should include age ranges. I think it may also be better to focus just on adult population studies rather than child, adult and older adults.

(4) My other concern is that this review is now outdated (end date was June 2023) as the authors mention there is a huge increase in literature regarding this topic. If this is the case then the review may need updating to include at least another year.

Reviewer #3: Please see attached comments. The major issue is that the period of review was from over 1.5 years ago, meaning by the time this is published, the most recent study referenced will be about two years old. In a rapidly growing field that is AI- and ML-based, this paper will be largely outdated at the time of publication.

Introduction

• It seems as though some more effort needs to be put into making sure the reader understands macro- and micro-adaptive strategies. The Figure (2) provided is not very intuitive for the macro-adaptive portion (a). Given that much of your argument and discussion revolves around a good understanding of these concepts, more care should be taken in driving home those points in the introduction.

Materials

• This paper has just now come to review, and the screening for articles was stopped in February 2023 (over 1.5 years ago now). Given the fast-moving state of this field, especially pertaining to use of AI methods, this manuscript search will likely need to be updated, as it seems some delay has occurred between the time you wrote this paper and when it was submitted for review. The only issue here is that when someone cites your review, if published in 2024, it will be referring to literature reviewed from nearly two years prior to the publication of this article.

Results

• Reframe the question, “How effective are they in empirical CT studies?” in 3.4 Q4 to be a complete, standalone sentence (i.e. don’t use the word “they”, as the reader does not know what you’re referring to).

• Grammar and punctuation issues with 3.5 Q5 (line 600; random capitalization, “have” instead of “has”; line 604 has a random use of the number “2” in a sentence instead of “two”, as would be appropriate) – ensure a grammar check is done throughout the manuscript

• I feel like the results section could benefit from a figure that demonstrates what approaches were used by the most effective studies (e.g. those with positive outcomes likely due to their chosen CT approach and those which had ++ validation). If the reader comes to this paper wanting to know how best to design their ML-based, individualized, CT intervention they should easily be able to scroll to your results section and know the essentials of the procedures used most successfully to date.

Discussion

• Once the last comment I made above in the results section has been done, it will make re-organizing your discussion section much easier.

Reviewer #4: This paper is very interesting, innovative and well written. The objectives, research questions and results are very clear.

There are some issues to be addressed:

1. The flowchart (fig.1) is unclear in some parts, e.g. in the sections ‘studies excluded’ and ‘studies sought for retrieval’ it would be helpful to specify the reasons. In the section ‘studies included’, 17 articles are reported, but it is not made clear that these are 19 studies. Please review and add these.

2. In the paragraph Sorting keys of AI techniques for content adapting to learner’s capabilities the authors report that AI techniques can be classified into four main families, yet they only list three of them. Please make this information consistent.

3. Please check that all acronyms in the paper are spelled out in full at least once, e.g. in the paragraph Evaluation of AI techniques the acronym NFT is not specified.

4. The table 5 only shows 9 out of 19 studies, no reason is specified. Please clarify this point.

6. PLOS authors have the option to publish the peer review history of their article (what does this mean?). If published, this will include your full peer review and any attached files.

Reviewer #1: No

Reviewer #2: No

Reviewer #3: No

Reviewer #4: **Yes: **Dr. Laura Camillo

---

## [Author Response · Author response to Decision Letter 1]

9 Dec 2024

The following has been directly referenced from the 'Response to Reviewers' document:

---

Reviewer #1, comment #1

I think the use of AI should be mentioned much earlier in the Introduction. As it currently reads, it is

not mentioned until page 6.

Our response #1.1

At the end of the first paragraph of the introduction, we have added sentences clearly stating that our

focus is on existing AI strategies that enable individualized cognitive training (see lines [30-41]).

Reviewer #1, comment #2

Table 2 that reviews the papers should include age ranges. I think it may also be better to focus just

on adult population studies rather than child, adult and older adults.

Our response #1.2

We have added the adult age ranges to Tables 2 and 3. While we understand the suggestion to focus

on a specific population, the goal of this review is to examine all existing strategies for individualization.

Due to the limited number of studies available, we included all population types to provide a comprehensive

overview of the current state of the art.

Reviewer #1, comment #3

My other concern is that this review is now outdated (end date was June 2023) as the authors mention

there is a huge increase in literature regarding this topic. If this is the case then the review may need

updating to include at least another year.

Our response #1.3

We fully agree with the reviewer’s concern. Unfortunately, this systematic review was submitted in

October 2023, and the peer-review process was delayed due to initial incorrect reviews (unrelated to this

paper) and a set of AI-generated reviews. Consequently, there was no additional time to search for, process,

and incorporate more recent studies. We have clearly stated in the methods section when the literature

search was conducted, ensuring that readers are informed about the search timeline.

---

Reviewer #2, comment #1

The major issue is that the period of review was from over 1.5 years ago, meaning by the time this is

published, the most recent study referenced will be about two years old. In a rapidly growing field that is

AI- and ML-based, this paper will be largely outdated at the time of publication.

Our response #2.1

Please refer to our response to Reviewer #1 for an explanation regarding the timeline and limitations

in updating the review.

Reviewer #2, comment #2

It seems as though some more effort needs to be put into making sure the reader understands macro- and

micro-adaptive strategies. The Figure (2) provided is not very intuitive for the macro-adaptive portion (a).

Given that much of your argument and discussion revolves around a good understanding of these concepts,

more care should be taken in driving home those points in the introduction.

Our response #2.2

We have revised the paragraph explaining micro- and macro-adaptive strategies to improve clarity (see

specifically section [224-255]). Additionally, Figure 2 has been updated by removing unnecessary elements,

such as the starting point square, to enhance its intuitiveness.

Reviewer #2, comment #3

This paper has just now come to review, and the screening for articles was stopped in February 2023

(over 1.5 years ago now). Given the fast-moving state of this field, especially pertaining to use of AI methods,

this manuscript search will likely need to be updated, as it seems some delay has occurred between the time

you wrote this paper and when it was submitted for review. The only issue here is that when someone cites

your review, if published in 2024, it will be referring to literature reviewed from nearly two years prior to

the publication of this article.

Our response #2.3

Please refer to our response to Reviewer #1 for an explanation regarding the timeline and limitations

in updating the review.

Reviewer #2, comment #4

Reframe the question, “How effective are they in empirical CT studies?” in 3.4 Q4 to be a complete,

standalone sentence (i.e. don’t use the word “they”, as the reader does not know what you’re referring to).

Our response #2.4

We have reformulated the question in section 3.4 Q4 for clarity (see [363-365]).

Reviewer #2, comment #5

Grammar and punctuation issues with 3.5 Q5 (line 600; random capitalization, “have” instead of “has”;

line 604 has a random use of the number “2” in a sentence instead of “two”, as would be appropriate) –

ensure a grammar check is done throughout the manuscript

Our response #2.5

We thank the reviewer for identifying these typos. A thorough grammar check has been completed

throughout the manuscript.

Reviewer #2, comment #6

I feel like the results section could benefit from a figure that demonstrates what approaches were used

by the most effective studies (e.g. those with positive outcomes likely due to their chosen CT approach and

those which had ++ validation). If the reader comes to this paper wanting to know how best to design their

ML-based, individualized, CT intervention they should easily be able to scroll to your results section and

know the essentials of the procedures used most successfully to date.

Our response #2.6

We have created a new figure that integrates various results, including the impact of training and methodological

quality, to highlight the approaches used by the most effective studies (see Figure 4: ”Comparison

of studies across single and multi-domain interventions in Cognitive Performance, Training Trajectories, and

Subjective Experience.”).

---

Reviewer #3, comment #1

The flowchart (fig.1) is unclear in some parts, e.g. in the sections ‘studies excluded’ and ‘studies sought

for retrieval’ it would be helpful to specify the reasons. In the section ‘studies included’, 17 articles are

reported, but it is not made clear that these are 19 studies. Please review and add these.

Our response #3.1

We have updated the flowchart to clarify the distinction between the number of articles included and

the number of studies reviewed, as some articles contain multiple studies. Additionally, we specified the

reasons for exclusions in the sections ‘studies excluded’ and ‘studies sought for retrieval’.

Reviewer #3, comment #2

In the paragraph Sorting keys of AI techniques for content adapting to learner’s capabilities the authors

report that AI techniques can be classified into four main families, yet they only list three of them. Please

make this information consistent.

Our response #3.2

We have revised this section to ensure consistency and clarity in the classification of AI techniques (see

lines [224-255]).

Reviewer #3, comment #3

Please check that all acronyms in the paper are spelled out in full at least once, e.g. in the paragraph

Evaluation of AI techniques the acronym NFT is not specified.

Our response #3.3

We conducted a thorough check to ensure all acronyms are defined at least once throughout the paper.

The acronym NFT was spelled out on first use in the “Introduction” (line 96).

Reviewer #3, comment #4

The table 5 only shows 9 out of 19 studies, no reason is specified. Please clarify this point.

Our response #3.4

Table 5 excludes studies with non-comparative designs, as these automatically receive the lowest grade.

We have added this clarification in the table description to ensure transparency

---

## [Editor Report · Decision Letter 1]

17 Dec 2024

Exploring the potential of artificial intelligence in individualized cognitive training: A systematic review

PONE-D-23-33514R1

Dear Dr. Adolphe,

We’re pleased to inform you that your manuscript has been judged scientifically suitable for publication and will be formally accepted for publication once it meets all outstanding technical requirements.

Kind regards,

Alessandro Bruno, Ph.D.

Academic Editor

PLOS ONE

Additional Editor Comments (optional):

Dear Authors,

I appreciate your efforts in punctually answering the reviewers' comments and remarks.

As far as I am concerned, your manuscript is now ready for acceptance.

My best regards,

A.B.
---

## [Editor Report · Acceptance letter]

PONE-D-23-33514R1

PLOS ONE

Dear Dr. Adolphe,

I'm pleased to inform you that your manuscript has been deemed suitable for publication in PLOS ONE. Congratulations! Your manuscript is now being handed over to our production team.

Kind regards,

on behalf of

Associate Professor Alessandro Bruno

Academic Editor

PLOS ONE